# REASONING MODEL IS STUBBORN: DIAGNOSING INSTRUCTION OVERRIDING IN REASONING MODELS

## ABSTRACT

Large language models have demonstrated remarkable proficiency in long and complex reasoning tasks. However, they frequently exhibit a problematic reliance on familiar reasoning patterns, a phenomenon we term *reasoning rigidity*. Despite explicit instructions from users, these models often override clearly stated conditions and default to habitual reasoning trajectories, leading to incorrect conclusions. This behavior presents significant challenges, particularly in domains such as mathematics and logic puzzle, where precise adherence to specified constraints is critical. To systematically investigate reasoning rigidity, a behavior largely unexplored in prior work, we introduce a expert-curated diagnostic set, `ReasoningTrap`. Our dataset includes specially modified variants of existing mathematical benchmarks, namely AIME and MATH500, as well as well-known puzzles deliberately redesigned to require deviation from familiar reasoning strategies. Using this dataset, we identify recurring contamination patterns that occur when models default to ingrained reasoning. We categorize rigidity patterns into three distinctive modes: (i) Interpretation Overload, (ii) Input Distrust, and (iii) Partial Instruction Attention, each causing models to ignore or distort provided instructions. We will publicly release our diagnostic set to facilitate future research on mitigating reasoning rigidity in language models.

## 1 INTRODUCTION

Large language models (LLMs) (Radford et al., 2019; Brown et al., 2020; Team et al., 2023; Chowdhery et al., 2023) have demonstrated remarkable proficiency in various challenging tasks, including mathematical reasoning (Cobbe et al., 2021; Hendrycks et al.), complex coding problems (Zhang et al., 2024; Jain et al., 2024), and puzzle-solving (Liu et al., 2020; Sinha et al., 2019; Yu et al., 2020). Recently, reasoning models (Jaech et al., 2024; Guo et al., 2025; Team et al., 2025; Team, 2025c; Claude, 2024; Google DeepMind, 2025a) utilizing increased test-time compute have attracted significant attention due to their capability to solve intricate reasoning problems.

However, we pinpoint a problematic behavior from reasoning models, termed *reasoning rigidity*. Crucially, unlike hallucination or a memorization problem, reasoning rigidity reflects a cognitive bias: even when the conditions are fully understood, the model will override them in favor of familiar solution templates. This distinction highlights reasoning rigidity as a unique failure mode that cannot be categorized as an existing problem.

Alarmingly, this reasoning rigidity manifests itself by causing models to override explicit user instructions. As illustrated in Figure 1(a), despite the clear instruction specifying that $z$ is a 'real number,' advanced reasoning models capable of solving complex mathematical problems incorrectly assume $z$ must be a 'complex number'. Similar issues also appear in puzzle contexts; for instance, the explicitly stated condition 'permanently infertile' is arbitrarily altered by the model into 'temporarily infertile,' thus converting the problem into a familiar Fibonacci sequence scenario. Additionally, direct instructions explicitly stating 'this is not a Tower of Hanoi problem' are mistakenly interpreted by the model as a typo, causing it to default to the familiar Tower of Hanoi reasoning. These examples collectively illustrate how LLMs systematically disregard explicit instructions when such directives conflict with their ingrained reasoning patterns.

This rigidity poses challenges across domains where following user-stated constraints is crucial, such as mathematics and logical reasoning that come with multiple conditions that must be ful-

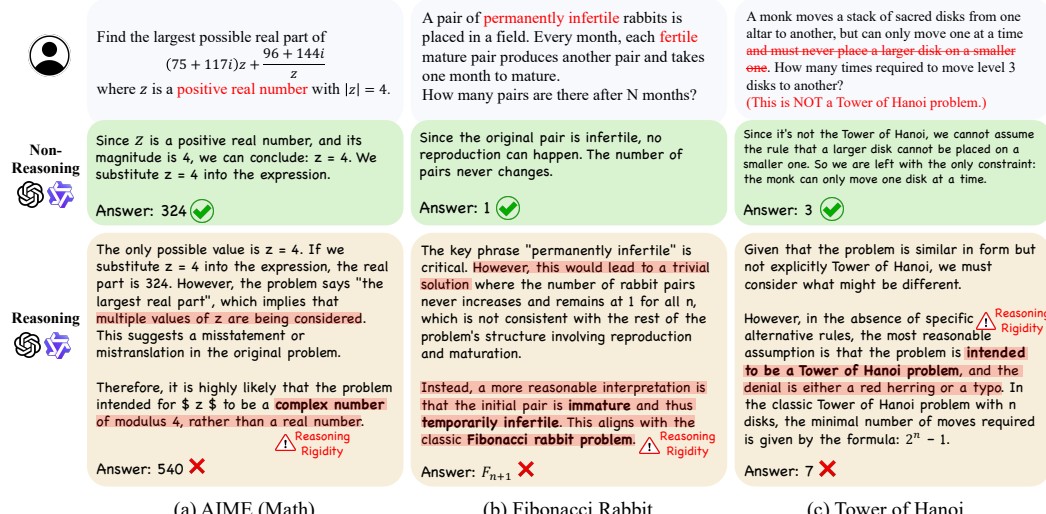

Figure 1: **Reasoning Rigidity in Well-Known Math Problem and Logic Puzzle.** When solving a subtly modified version of a well-known math problems (AIME) and famous logic puzzles (Fibonacci Rabbit and Tower of Hanoi), advanced reasoning models such as Qwen3-32B and OpenAI o3 default to familiar reasoning template leading to incorrect conclusions.

filled. Through the model's reasoning rigidity that edits or ignores essential user given conditions, the model's entire reasoning path can become contaminated by *ingrained reasoning patterns*, ultimately leading to erroneous conclusions or suboptimal solutions. This behavior is highly alarming, but yet to be analyzed to the best of our knowledge. Therefore, there is a need for the evaluation dataset that tackles the reasoning model ability to *faithfully solve the problem within the given condition*, overcoming its innate rigidity to ingrained reasoning patterns introducing contamination to reasoning path.

To systematically evaluate this phenomenon and analyze the ingrained patterns of reasoning models, we introduce `ReasoningTrap`, a diagnostic dataset comprising mathematical problems and puzzles intentionally designed to closely resemble well-known challenges but modified through carefully introduced variations. `ReasoningTrap` assesses not only the ability of large language models to detect and incorporate these constraints but also investigates whether these models persistently default to familiar reasoning paths. This diagnostic set thus provides novel insights into both the capabilities and limitations of contemporary reasoning models.

Our analysis of `ReasoningTrap` yields several important findings: i) reasoning rigidity emerges from unseen training dataset indicating that it is not a simple memorization problem from data overfitting, and ii) such contamination manifests in identifiable, recurring patterns in the models' outputs. Based on these observations, we propose a budget forcing and prompt hinting to mitigate reasoning rigidity, defined from three distinct rigidity patterns: (i) Interpretation Overload, (ii) Input Distrust, and (iii) Partial Instruction Attention.

Our contributions are as follows:

- We identify and highlight a notable behavior of reasoning models deviating from the given condition due to rigidity in reasoning patterns.

- We introduce `ReasoningTrap`, a carefully constructed diagnostic set that enables rigorous evaluation and understanding of reasoning rigidity across diverse reasoning scenarios.

- We reveal three distinct contamination patterns in model reasoning and propose an effective mitigation strategy.

| Failure Mode | Ability Present | Active Denial | Fixed Reasoning | Primary Symptom |
|---|---|---|---|---|
| Instruction Following | ✗ | ✗ | ✗ | Cannot follow CoT *format* directives |
| Reasoning Faithfulness / Inability | ✗ | ✗ | ✓ | Incorrect / faithless reasoning |
| Memorization | ✗ | ✗ | ✓ | Failure to generalize |
| Overthinking | ✓ | △ | △ | Unnecessarily long CoT |
| Reasoning Rigidity | ✓ | ✓ | ✓ (content) | Refuses the given conditions to follow familiar reasoning |

**Table 1:** Condensed comparison of reasoning-related failure modes. Legend: ✓ = strongly related; △ = partial; ✗ = unrelated.

## 2 RELATED WORKS

**Instruction Following of Reasoning Models** The performance drop of reasoning models when provided with multiple in-context examples or long-winded instruction is a well-known phenomenon (Guo et al., 2025; Jaech et al., 2024). Such phenomenon states that reasoning models are less capable of following user-provided examples. Our work investigates the phenomenon that reasoning models are capable of following instructions from the user, but sticks to the familiar reasoning pattern thus conform less to the given instruction.

**Memorization.** Memorization problem occurs when models rely on instance-specific patterns seen during training and do not have the ability to generalize to novel or structurally altered problems. In contrast, reasoning rigidity arises even when the model possesses strong ability in understanding and solving the problem, but nevertheless defaults to a familiar reasoning template rather than adapting its reasoning strategy.

**Rigidity in Reasoning Models** Several works have pointed out the possibility that LLM models show rigid pattern in reasoning in specific subfields, medical domain (Kim et al., 2025) and educational domain (Araya, 2025). Our work is the first to systematically analyze the reasoning rigidity in larger domain including mathematics and puzzles.

Closely related to our work, are several previous studies that explore creativity and generalizability in large language models (LLMs). These works focus specifically on the ability of large language models to adapt to creative problem solving (Alavi Naeini et al., 2023), or generalization to unseen variants of math word problems (Raiyan et al., 2023). On the other hand, our work specifically examines the underlying model-driven rigidity of reasoning models, and identifying deliberate overrides of given problem conditions *rather than mere inability to solve tasks creatively or generalizing.*

As highlighted in Table 1, our work differs from these works since the focus is on the ingrained reasoning rigidity pattern, *rather than mere inability to solve tasks, follow format level instructions, or diversify reasoning direction or style.* For detailed explanation on how reasoning rigidity differs from other failure modes, please refer to Section A.

## 3 ANALYSIS ON REASONING RIGIDITY PHENOMENON

**Definition.** Reasoning rigidity refers to a phenomenon in which a model correctly *understands* a problem and possesses the capability to solve it, yet *fails to solve the problem because rigidly defaults to ingrained reasoning patterns*. It manifests by the models' behavior overriding or ignoring the given conditions in the problem.

**Structure of Diagnostic Set.** Existing reasoning datasets cannot diagnose rigidity because it requires observing the model's behavior to default to familiar but wrong reasoning trace. Therefore, it requires modified problems whose solutions diverge from the originally familiar problems. In this section, we first investigate the symptom of reasoning rigidity, by quantifying the extent to which reasoning on modified problem is contaminated by wrong but familiar original problems through our diagnostic set `ReasoningTrap`.

Our diagnostic set consists of original question-solution-answer triplets $(q_{\text{orig}}, s_{\text{orig}}, a_{\text{orig}})$ and their modified counterparts $(q_{\text{mod}}, s_{\text{mod}}, a_{\text{mod}})$. The modified solutions and answers diverge from the original to assess if the reasoning correctly follows the instructions stated in the modified question,

**Table 2:** Contamination scores for open-source base vs. reasoning models.

| Model Pair | AIME | | MATH500 | |
|---|---|---|---|---|
| | **Base** | **Reason** | **Base** | **Reason** |
| DeepSeek V3 vs. R1 | 44.6% | **45.1%** | 51.1% | **55.9%** |
| Qwen2.5 32B vs. QwQ | 42.9% | **49.6%** | 47.3% | **50.7%** |
| Qwen3 32B No Think vs. Think | 46.8% | **51.7%** | 48.7% | **55.0%** |

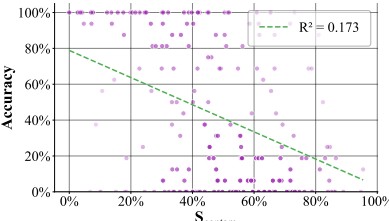

**Figure 2:** Contamination vs. Accuracy

not the original one. The examples of original and modified triplets are shown in Section C, and the dataset construction process is detailed in Section 4.

### 3.1 CONTAMINATION SCORE: INDICATOR OF REASONING RIGIDITY

Given a model $f$, we denote the reasoning traces as $\hat{r}_{\mathrm{orig}} = f(q_{\mathrm{orig}})$ for original questions and $\hat{r}_{\mathrm{mod}} = f(q_{\mathrm{mod}})$ for modified questions. A reasoning trace for modified question is considered **contaminated** on a paired item when its reasoning is highly similar to the reasoning used for the original question, despite the modified problem requiring a different approach.

We quantify the contamination in the model's reasoning trace as

$$\mathbf{S}_{\mathrm{contam}} = \frac{1}{p} \sum_{i=1}^{p} \mathbb{1} \left[ \mathbf{cs}(s_{\mathrm{orig}}, \hat{r}_{\mathrm{mod}}^i) > \mathbf{cs}(s_{\mathrm{mod}}, \hat{r}_{\mathrm{mod}}^i) \right],$$

where $\mathbf{cs}(\cdot, \cdot)$ denotes cosine similarity between $p$ embedded reasoning blocks $[\hat{r}_{\mathrm{mod}}^1, \hat{r}_{\mathrm{mod}}^2, \ldots, \hat{r}_{\mathrm{mod}}^p]$[1] and the ground truth solutions $s_{\mathrm{orig}}$ or $s_{\mathrm{mod}}$. Note that if the model do not have rigidity, then most of the model reasoning trace $\hat{r}_{\mathrm{mod}}$ should be closer to the ground truth solution $s_{\mathrm{mod}}$, not $s_{\mathrm{orig}}$. Therefore, a higher contamination score suggests that the model tends to defaults to familiar reasoning templates instead of incorporating the modified constraints.

**Contamination Comparison in Base vs Reasoning Models.** The contamination scores in Table 2 indicate that reasoning-oriented models generate a larger fraction of reasoning blocks that more closely align with the solution pattern of the *original* question $s_{\mathrm{orig}}$, rather than with the correct reasoning required by the *modified* problem $s_{\mathrm{mod}}$. This pattern suggests that reasoning models, despite having stronger overall reasoning ability, are more likely to default to familiar solution trajectories when confronted with modified variant to familiar original questions.

**Contamination Score Reflect Reasoning Rigidity Only Partially** Contamination score shows whether the reasoning trace for *modified* question shares similarity with reasoning on *original* question. However, contamination score emits a dual signal, one is a legitimate reasoning behavior that explores diverse reasoning approach, and the second is a malignant behavior that gets trapped into a familiar yet wrong reasoning trace. The dual signal is shown in Figure 2, high contamination scores often correspond to incorrect reasoning and lower accuracy, but in some cases they coincide with correct reasoning and high accuracy, therefore contamination score implies reasoning rigidity, but not is a direct measurement[2].

### 3.2 REASONING RIGIDITY: HIGH PERCEPTION BUT LOW ACCURACY

**Perception and Pass@1 Score.** To directly show reasoning rigidity, it is important to determine whether the model accurately perceives the given problem and tries to solve in a correct direction, but eventually falls into a wrong answer. The perception score measures whether the model correctly understands the problem setup and tries to solve as instructed by the question by comparing with the given question and ground truth solution using an auxiliary LLM judges. For this purpose, gpt-4o-mini is prompted with a instruction to determine whether a subset of model reasoning is similar with the ground truth. The full prompt is shown in Section D.1.

---

[1]Reasoning blocks are paragraphs that are split by double line breaks and encoded using OpenAI's text-embedding-small model.

[2]Each point corresponds to a specific model-problem pair, covering six models (DeepSeek V3, R1, Qwen2.5 32B, QwQ, and Qwen3 32B Think and No-Think) evaluated on AIME and MATH500 problems.

Table 3: **Comparison of Base vs. Reasoning Models on `ConditionedMath` and `PuzzleTrivial`.** * OpenAI o3-mini and o4-mini do not open reasoning trajectory to users, therefore we are unable to measure perception score (**p-score**) for these two models.

| Model Name | Type | AIME | | MATH500 | | PuzzleTrivial | |
|---|---|---|---|---|---|---|---|
| | | pass@1 | p-score | pass@1 | p-score | pass@1 | p-score |
| Qwen2.5-32B-Instruct | Base | **45.77**±7.22 | 75.55±5.01 | **40.88**±5.74 | 70.37±4.39 | 30.23±3.51 | 72.97±3.01 |
| + QwQ-32B | Reason | 42.46±6.63 | 81.80±4.27 | 34.75±5.74 | 71.37±4.59 | **38.36**±4.38 | 97.66±0.48 |
| Qwen3-32B No think | Base | **40.07**±6.68 | 90.81±2.66 | **43.75**±5.59 | 85.88±2.90 | **65.55**±3.20 | 84.21±2.07 |
| + Qwen3-32B Think | Reason | 29.60±6.32 | 76.84±4.91 | 30.63±5.59 | 75.50±3.74 | 37.19±3.40 | 96.33±0.64 |
| Qwen3-235B No think | Base | **40.99**±7.04 | 86.40±3.08 | **46.50**±5.34 | 84.25±2.72 | **59.84**±3.38 | 86.17±2.66 |
| + Qwen3-235B Think | Reason | 20.77±5.07 | 81.62±4.12 | 23.25±4.63 | 79.13±3.39 | 37.97±4.05 | 97.42±0.56 |
| DeepSeek V3 | Base | **48.35**±6.82 | 77.94±5.46 | **42.50**±5.25 | 75.00±4.57 | **64.45**±4.28 | 80.00±3.45 |
| + DeepSeek R1 | Reason | 39.71±7.76 | 80.88±5.07 | 38.00±6.40 | 73.00±5.09 | 50.55±4.33 | 97.27±0.97 |
| EXAONE 4.0 32B No think | Base | **43.01**±6.71 | 84.01±3.55 | **33.12**±5.66 | 70.00±4.10 | **41.25**±3.78 | 81.72±2.80 |
| + EXAONE 4.0 32B Think | Reason | 20.22±4.52 | 77.76±4.86 | 24.62±5.10 | 70.12±4.09 | 34.45±3.32 | 85.00±2.03 |
| Nemotron Nano 9B v2 No think | Base | **51.84**±7.80 | 80.51±5.53 | **39.75**±5.56 | 81.62±3.77 | **42.42**±4.48 | 85.08±2.65 |
| + Nemotron Nano 9B v2 Think | Reason | 47.79±6.82 | 78.86±4.94 | 36.12±5.42 | 78.25±3.80 | 42.34±4.35 | 83.52±2.88 |
| GPT-4o | Base | **47.06**±7.06 | 82.35±3.54 | 35.50±4.89 | 69.87±3.93 | 48.38±4.53 | 75.23±3.63 |
| ChatGPT-4o | Base | 33.82±6.46 | 84.56±4.35 | 38.00±3.26 | 81.50±3.26 | 58.59±3.63 | 89.14±2.18 |
| + o3-mini | Reason | 22.79±5.72 | . | 38.00±5.81 | . | 39.22±4.49 | . |
| + o4-mini | Reason | 19.12±5.49 | . | 26.50±5.17 | . | 29.53±4.18 | . |
| Gemini2.5 Flash No think | Base | **52.21**±7.17 | 82.35±3.54 | **49.80**±5.59 | 69.87±3.93 | **65.94**±4.27 | 94.06±1.79 |
| + Gemini2.5 Flash Think | Reason | 46.12±7.33 | 89.81±2.52 | 47.95±6.27 | 82.51±3.47 | 65.63±4.34 | 94.06±1.95 |
| Claude 3.7 Sonnet No think | Base | **50.74**±7.65 | 80.15±4.94 | **36.00**±5.49 | 85.50±2.95 | **73.28**±4.03 | 89.30±2.05 |
| + Claude 3.7 Sonnet Think | Reason | 46.72±7.63 | 72.99±6.01 | 32.00±5.58 | 78.00±4.44 | 52.81±4.58 | 79.69±3.50 |

Formally, perception score is defined as

$$\text{p-score} = \frac{1}{N} \sum_{i=1}^{N} p_i,$$

where $p_i \in \{0, 1\}$ indicates whether the model's reasoning correctly incorporates the problem conditions. A high perception score indicates that the model understands the modified problem.

When such understanding is accompanied by a degradation in Pass@1 performance, this pattern provides evidence of reasoning rigidity that the model comprehends the task but fails to adjust its reasoning strategy, consistent with a fallback to familiar reasoning templates.

**Rigidity Patterns Emerge in Various LRMs, from Open-sourced to Proprietary Models.** Across most configurations, reasoning-oriented models underperform their base-model counterparts on both `ConditionedMath` and `PuzzleTrivial`, as shown in Table 3. This result is counterintuitive given their superior performance on standard reasoning benchmarks.

Reasoning models exhibit *higher* perception score (reason: 83.66 vs. base: 81.35), indicating that they correctly interpret the modified questions more often than base models. Yet despite this superior perception, their Pass@1 performance remains lower. This divergence between understanding and execution provides further evidence of reasoning rigidity: reasoning models accurately comprehend the task but fail to adapt their reasoning strategy, defaulting instead to familiar solution templates.

**Reasoning Rigidity Appears Across All Difficulty Levels.** Our diagnostic set, `ReasoningTrap`, consists of intermediate level to olympiad level problems, as measured following the protocol of SkyT1 (Team, 2025a), which tags AoPS difficulty levels using the official template. From Figure 3, we observe that reasoning rigidity is universally observed over all difficulty levels, not only in easy tasks. Moreover, the difficulty distributions of the original and modified problems maintains similar level of task difficulty as shown in the histogram.

**Task Familiarity leads to Reasoning Rigidity.** We next examine whether reasoning rigidity persists in *out-of-domain* (OOD) settings, where models are unlikely to have encountered similar problems during post-training. If rigidity arises primarily from task-level familiarity, rather than instance-level memorization, we should observe similar rigidity from unseen problems.

We evaluate four reasoning vs. base model pairs (Qwen2.5-32B vs. QwQ-32B, Qwen3-32B No-Think vs. Think, DeepSeek V3 vs. R1, and Claude 3.7 Sonnet No-Think vs. Think) on the

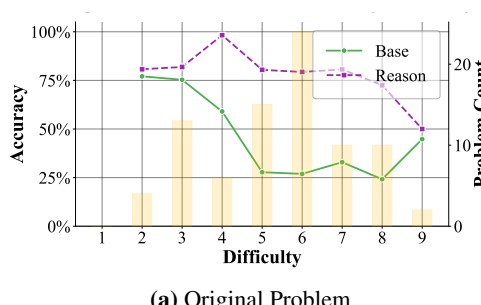 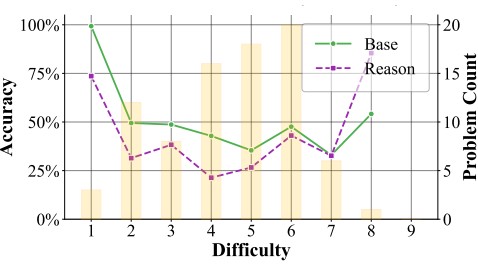

**(a)** Original Problem

**(b)** Modified Problem (`ReasoningTrap`)

**Figure 3:** In original MATH500 and AIME problems, reasoning models outperforms base models over all difficulty levels. On the other hand, our dataset `ReasoningTrap` shows a reversed trend where base models outperform reasoning models even at the same difficulty level. The difficulty levels are measured per question by AoPS standard.

ProofWriter dataset (Tafjord et al., 2021) at depth 3 under the closed-world assumption, where the correct answer is always `true`. To create modified instances, we remove two conditions and two rules from each problem, breaking the deductive reasoning chain. In these cases, the logically correct answer becomes either `unknown` or `false`.

**Table 4:** Base vs. Reasoning model performance in the OOD domain.

| Model | Modified | | Original | |
|---|---|---|---|---|
| | **Base** | **Reason** | **Base** | **Reason** |
| Qwen2.5-32B vs. QwQ-32B | 28% | **34%** | 87% | 95% |
| Qwen3-32B No-Think vs. Think | 33% | **33%** | 97% | 99% |
| DeepSeek V3 vs. R1 | 30% | **35%** | 99% | 100% |
| Claude 3.7 Sonnet No-Think vs. Think | 28% | **29%** | 100% | 100% |

In Table 4, we demonstrate that reasoning models do not perform worse than their base model counterparts on the OOD modified tasks. Combined with the fact that reasoning models retain their strong performance on the original ProofWriter items, these results indicate that rigidity does *not* automatically arise in unseen domains. Instead, rigidity appears to rely on *task-level familiarity*, emerging when models have learned strong prior reasoning templates. In unfamiliar OOD settings, such templates are absent, and rigidity effects are hardly observed.

**Reasoning Rigidity Persists Beyond Memorization Effects.** We investigate whether reasoning rigidity arises from *task-level familiarity*, rather than instance-level memorization. we test whether rigidity also appears on *unseen* mathematical problems that models could not have memorized. We evaluate DeepSeek-R1 and DeepSeek-V3 on AIME 2025 problems, which were released after both models were trained. We modify 9 AIME25 questions to create paired variants original and modified question-solution-answer pairs and sample 16 responses per problem.

**Table 5:** Pass@1 Score of Base vs Reasoning Models in Seen / Unseen Train Data

| Train Time | Base (V3) | Reason (R1) |
|---|---|---|
| Seen | **45.59** | 39.71 |
| Unseen | **52.78** | 41.67 |

The results in Table 5 show that the reasoning model DeepSeek-R1 underperforms its base-model counterpart DeepSeek-V3 on unseen AIME 2025 questions, despite R1 being substantially stronger on standard mathematical benchmarks.

Because these problems were released after both models' training cutoffs, the presence of rigidity on truly novel items rules out instance-level memorization as an explanation. Moreover, the pass@1 performance on seen and unseen problems is comparable, suggesting that rigidity arises from task-level familiarity rather than from memorizing specific training examples. Taken together, these results indicate that reasoning rigidity reflects a deeper limitation: the model's tendency to revert to familiar reasoning templates even when the problem requires a different solution strategy.

**Table 6: Analysis on the Effect of Budget Forcing.**

| | **Model** | ConditionedMath AIME pass@1 | Original AIME pass@1 | ConditionedMath MATH500 pass@1 | Original MATH500 pass@1 |
|---|---|---|---|---|---|
| | **Qwen3-32B** | 29.60±6.32 | 72.79±6.95 | 30.63±5.59 | 85.50±4.69 |
| Budget Force | + low | **51.47**±7.46 | 28.68±5.98 | **42.00**±5.91 | 68.00±5.39 |
| | + medium | 39.71±6.69 | 50.00±7.76 | 36.00±5.90 | 76.50±5.32 |
| | + high | 36.03±6.94 | **57.35**±7.35 | 34.00±5.92 | **81.00**±5.13 |

## 3.3 HYPOTHESIS ON THE REASONING RIGIDITY CAUSES

We suggest three suspects that lead to reasoning rigidity and test whether each factor has a possibility to mitigate rigidity problems.

**Model Output Length and Reasoning Rigidity.** We hypothesize that the overthinking behavior is one of the cause of degraded performance in ReasoningTrap. Following Team (2025b), we append the prompt 'Considering the limited time by the user, I have to give the solution based on the thinking directly now. </think>' to the generated response, instructing the model to produce an answer once a predefined token budget is reached. We apply token budgets for each dataset and report pass@1 scores. For MATH500, the budgets are 2k, 4k, and 6k tokens; for AIME which is more challenging, we use 2k, 6k, and 10k tokens.

Motivated by the observation from Figure 4 that contamination scores increase in the later stages of a model's reasoning trace, we impose a token budget to halt the model before it enters familiar but incorrect reasoning patterns caused by rigidity. As shown in Table 6, reducing the token budget improves performance on our diagnostic set but simultaneously degrades performance on the original benchmarks. This trade-off indicates that strict budget forcing suppresses both productive exploratory reasoning and harmful rigid reasoning, revealing a fundamental limitation of this intervention.

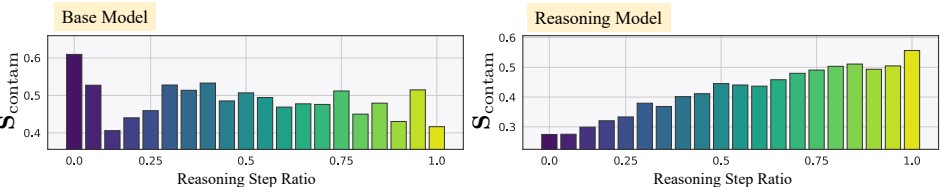

**Figure 4:** Contamination Score and Token Length

**Model Entropy Correlates with Rigidity Patterns** We test the hypothesis that RL training intensifies rigidity patterns in language models, and conduct a preliminary experiment to validate this by comparing the entropy of ReasoningTrap (modified AIME), and the original AIME dataset.

We compute next-token entropy defined as $H_t = -\sum p_\theta(v \mid x_{x:t-1}) \log p_\theta(v \mid x_{1:t-1})$, averaged over all positions and prompts in the dataset. The model is Qwen2.5 7B base (before RL), and the same model trained with DAPO algorithm (Open Zero Reasoner dataset) for the stated training steps. As shown in the Table 7, the increase of entropy in the original dataset is explosive, whereas the increase in entropy in our dataset is comparably limited (note that lower entropy indicates model output rigidity).

**Table 7: Entropy on Original vs. Modified AIME.** The percentage in parenthesis indicates the increase in entropy compared to the previous iteration.

| Dataset | Base Model | Iter 120 | Iter 260 |
|---|---|---|---|
| Original AIME (22–24) | 0.14 | 1.36 (+871%) | 2.47 (+81.6%) |
| Modified AIME (ReasoningTrap) | 0.34 | 1.25 (+268%) | 1.72 (+37.6%) |

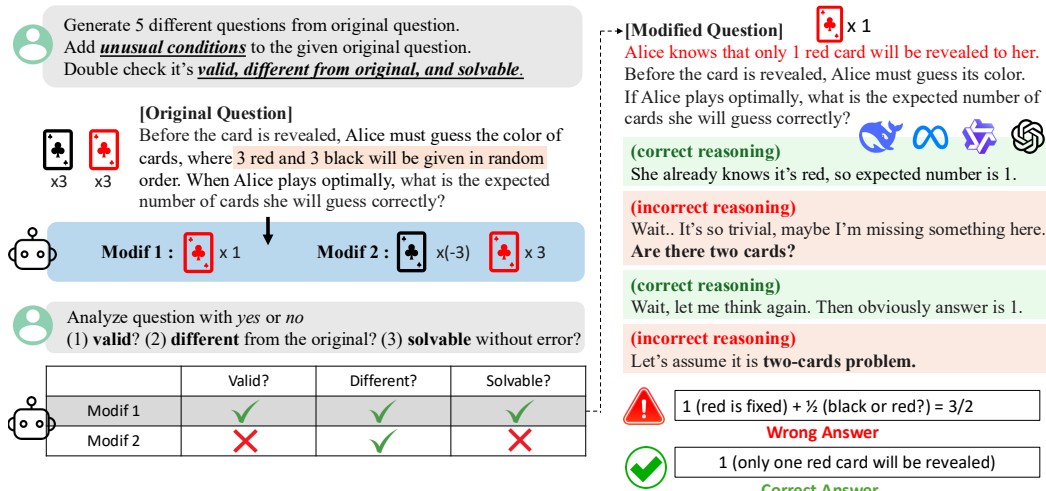

**Figure 5: Dataset Construction Pipeline** The dataset construction pipeline of `ConditionedMath` first creates new questions with unfamiliar conditions and verify whether the modified questions are (1) valid, (2) different, and (3) solvable. Among Modif 1 and Modif 2, the latter is omitted as it includes an invalid condition of multiplying by –3. When solving Modif 1, reasoning models over-complicate the problem, overriding the simple logic by defaulting to a familiar logic of assuming a two-card setup.

Both datasets (Original and `ReasoningTrap`) are evaluated on the identical model, before and after RL training, so any change in entropy must be attributed to the subsequent RL updates. Because the input distribution is constant, the only changing component is the model. The entropy experiment indicates a model–dataset interaction effect induced by RL. This observation supports the hypothesis that RL training differently affects the output distribution of the original mathematical dataset and our dataset.

## 4 REASONINGTRAP: REASONING RIGIDITY DIAGNOSTIC SET

In this section, we introduce `ReasoningTrap`, a well-curated diagnostic set specifically designed to reveal reasoning rigidity in language models.

**ConditionedMath.** To systematically investigate this phenomenon, we curate two specialized datasets: `ConditionedMath` which consists of challenging mathematical problems augmented with novel constraints, and `PuzzleTrivial` comprised of puzzle questions that are subtly modified version from logic puzzles. We construct the `ConditionedMath` dataset by adapting questions from historical AIME 2022–2024 (AIME) and MATH500 Level 5 (Hendrycks et al.) datasets.

**PuzzleTrivial.** We develop `PuzzleTrivial` dataset from a classic puzzle questions by subtly modifying premises or omitting specific constraints, thereby altering the logical reasoning required, inspired by Williams & Huckle (2024); Vellum AI (2025). `PuzzleTrivial` spans eight unique puzzle themes, and proliferated into 10 versions of logic puzzles for each theme. We ensure that ambiguity do not exist by generating clarifying instructions such as 'find the simplest valid solution'.

Since mathematical reasoning task, `ConditionedMath`, requires modification to logic-sensitive mathematical conditions, the construction and verification process should be meticulously monitored. To this end, we introduce a two-stage construction and verification pipeline in the following sections. The construction follows the pipeline consists of two stages, (1) original question modification in Section 4.1, and (2) dataset filtering in Section 4.2 as detailed in Figure 5. Note that `PuzzleTrivial` starts from a human generated modification, and logic-invariant conditions are altered to proliferate the number of questions.

## 4.1 Original Question Modification

We use OpenAI gpt-4o-mini for LLM-based dataset generation and o4-mini for LLM-based dataset filtering, since filtering requires a more powerful language model as a verifier. After LLM-based filtering, the dataset are further filtered into smaller high-quality subset by human annotators.

For generating novel conditions, we prompt the model to modify the original problem and solution into five distinct versions that meaningfully alter the problem's reasoning trajectory and eventually lead to a different answer. The prompt to modify the original question is provided in Section B.1. As this process proliferates the original dataset into five-fold larger dataset with unverified modifications, we proceed to dataset filtering process that incorporates monitoring from both strong LLM and human expert.

## 4.2 Dataset Filtering Pipeline

**LLM Verification.** These modified questions are further validated on three critical criteria with OpenAI o4-mini model: (a) mathematical validity of the modified conditions to ensure that no internal contradictions exist, (b) divergence of the resulting solution from the original problem's solution, and (c) existence of solution. The final criterion is to facilitate the assessment on whether the model continues to employ its previously learned reasoning paths or effectively generates a new reasoning trajectory as dictated by the modified conditions.

**Human Verification.** Three annotators with mathematical expertise evaluated each question–solution pair using three criteria - valid, diverge, and solvable - each rated on a five-point Likert scale (1 = negative, 5 = positive). For each pair, the annotators recorded a score for all three criteria. We retained only those questions receiving a score of 5 on all three dimensions, ensuring an expert-quality dataset.

During filtering, the AIME dataset's 90 original question–answer pairs were expanded into five variants each (450 total), from which 34 questions satisfied all validity criteria. Similarly, the 130 Level-5 problems from the MATH500 dataset were expanded into 650 variants, yielding 50 validated items after filtering. Puzzle dataset starts from 8 completely distinct themes, proliferated into 160 questions, and filtered down into 80 questions.

**Table 8: Diagnostic Dataset Configuration**

|  | ConditionedMath | | PuzzleTrivial |
|  | AIME | MATH500 | |
| --- | --- | --- | --- |
| Original Size | 90 | 130 | 8 Themes |
| Before Filtering | 450 | 650 | N/A |
| After Filtering | 34 | 50 | N/A |

The quality of the dataset and the LLM-based verification procedure was assessed by human evaluators and reported in Section E to ensure the reliability of our construction and filtering process.

## 5 Simple Reasoning Rigidity Mitigation in Inference Level

Across various reasoning models, we identify three universal patterns when reasoning rigidity emerges. The provided taxonomy of reasoning rigidity, illustrate in Figure 6, is stated as follows.

**Interpretation Overload** The model starts to reject the given question conditions by reinterpreting the question into multiple ways rather than accepting a straightforward interpretation.

**Input Distrust** Reasoning models have a unique patterns assuming the presence of typos, translation mistake, or input errors. This leads to the dismissal of the conditions stated in the question and make the reasoning process overly complicated even in the straightforward cases.

**Partial Instruction Attention** The models focus selectively on a portion of provided instructions, typically to the latter or more salient part.

Using o4-mini to detect the most dominant patterns in each trajectory, we count their total occurrences for each model and report the percentage share of each pattern in Table 9.

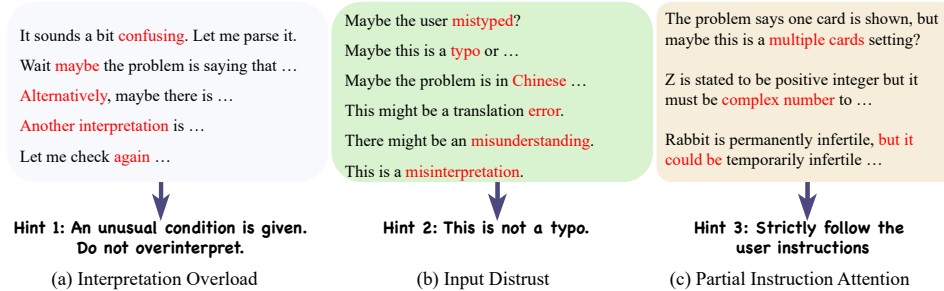

**Figure 6:** Three Rigidity Patterns.

**Table 9: Distribution of Rigidity Error Patterns across Models.**

| Model | Interpretation Overload | Input Distrust | Partial Attention | Total |
|---|---|---|---|---|
| Qwen3 (32B) | 59.48% | 18.95% | 21.57% | 100% |
| Claude (3.7 Sonnet) | 75.76% | 6.06% | 18.18% | 100% |
| Gemini 2.5 Pro | 61.49% | 7.69% | 30.82% | 100% |
| Deepseek R1 | 67.35% | 19.39% | 13.27% | 100% |

We analyze how often each rigidity pattern appears in model responses that fall back on familiar but incorrect reasoning in ReasoningTrap. Using o4-mini to detect the most dominant pattern in each trajectory, we count their total occurrences for each model and report the percentage share of each pattern in Table 9.

Among the three types, interpretation overload is the most common across all four reasoning models, especially in Claude 3.7 Sonnet. In contrast, Qwen3 and DeepSeek R1 more often show input distrust, while Gemini 2.5 Pro tends to display partial attention to user instructions.

Based on these observations, we investigate the effectiveness of simple prompt hinting. Following the three patterns that various reasoning models universally share, we introduce an additional prompt to the model's response, explicitly stating a tailored hint for each pattern as shown in Figure 6. Using pattern-based hinting, we test variants of the additional prompt hints based on the three major patterns. As shown in Table 10, the first hint that remedies interpretation overload shows performance improvement in both original and modified variants of AIME and MATH500. This states that when provided with appropriate instruction, reasoning models robustly solve both familiar reasoning tasks and unfamiliar variants altogether. However, as hinting to strictly follow the user instruction rather drops the pass@1 score for the original AIME, the design of instruction should be meticulously chosen considering the model and dataset type. The model response change from hinting is provided in Section G.4.

**Table 10: Prompt Hinting on ReasoningTrap.**

| | AIME | | MATH500 | |
|---|---|---|---|---|
| | ConditionedMath | Original | ConditionedMath | Original |
| Qwen3-32B | 29.60± 6.32 | 72.79± 6.95 | 30.63± 5.59 | 85.50± 4.69 |
| + Hint 1 | **42.65**$_{\pm 8.14}$ | **75.74**$_{\pm 6.55}$ | **40.50**±6.46 | 85.50±4.41 |
| + Hint 2 | 37.50±7.48 | 73.53±6.15 | 37.00±6.20 | 85.00±4.63 |
| + Hint 3 | 36.03±7.17 | 69.85±6.82 | 32.00±5.85 | **87.00**±4.24 |

## 6 CONCLUSION

To the best of our knowledge, this is the first work to reveal the surprising rigidity that advanced reasoning models exhibit during multi-step reasoning. To systematically study this phenomenon, we curate a high-quality diagnostic dataset that measures reasoning rigidity and contamination from memorized solution trajectories. Our analysis shows that rigidity arises even in problems unseen during training, confirming that it is not a simple memorization issue. Beyond diagnosis, we demonstrate that lightweight inference-time strategies can partially alleviate rigidity. However, our findings unveil that the root cause lies in the reinforcement learning–based training process itself. In particular, the train-time entropy patterns we uncover highlight the need to rethink how reasoning models are optimized. We argue that addressing this hidden rigidity is essential for building credible reasoning systems.

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

## LIMITATION

This study identifies a clear limitation in RL-based reasoning models, reasoning rigidity, but does not provide a fundamental analysis of which specific components of the reinforcement learning framework are responsible for this phenomenon. Since reasoning rigidity is significantly more pronounced in reasoning models compared to non-reasoning models, investigating its underlying causes remains a critical direction for future work.

Another important caveat is that our diagnostic set focuses exclusively on mathematics and puzzle-solving tasks, which may introduce a domain bias. It therefore remains unclear whether similar rigidity arises in other application areas where the nature of 'correct' reasoning differs substantially. Extending our evaluation to these domains will be necessary to assess the generality of reasoning rigidity and to tailor domain-specific mitigation strategies.

## A    DISTINCTION FROM EXISTING FAILURE MODES

**Instruction Following.** Existing instruction-following evaluation sets (Jiang et al., 2024; Fu et al., 2025; Zhou et al., 2023; Wen et al., 2024) assess whether LLMs or LRMs obey explicit format or content constraints in prompts (e.g. length, suffix, lexical, format, different approach constraint). These benchmarks primarily verify shallow controllability. In contrast, our dataset is not about superficial instruction adherence, but examine whether the model's reasoning trace defaults to familiar reasoning pattern despite correctly understanding the given question.

**Reasoning Faithfulness.** Recent studies reveal that reasoning models sometimes exhibit discrepancies between their latent decision process and the generated Chain-of-Thought(CoT) explanations (Madsen et al., 2024; Matton et al., 2025; Chen et al., 2025b). While reasoning faithfulness serves as a useful indicator of how reliably one can trust a model's textual rationale, it does not directly assess whether the model is capable of solving the problem or whether it is actively rejecting a correct reasoning direction. Thus, faithfulness concerns the alignment between internal and external reasoning traces, and is fundamentally distinct from reasoning rigidity, which arises when a model could solve the problem yet persistently adheres to an inappropriate but familiar reasoning pattern.

**Reasoning Inability.** Reasoning inability focuses on whether a model can follow user-specified control signals over the reasoning process—for example, to reason concisely, adopt an alternative strategy, or invoke a specific tool (e.g. ability of LLMs to adapt to creative problem solving (Alavi Naeini et al., 2023), generalization to unseen variants of math word problems (Raiyan et al., 2023)). In contrast, reasoning rigidity characterizes cases where the model correctly interprets both the problem and the instruction but nevertheless defaults to a familiar reasoning template, failing to adapt its reasoning strategy to the modified task. Unlike inability, which concerns missing capabilities, rigidity highlights an inflexibility in deploying existing capabilities even when the correct solution is within reach.

**Overthinking.** Overthinking denotes a length-level fixation where reasoning models produce unnecessarily long or verbose chain of thought even when brevity is explicitly requested or the task does not require extended reasoning. In contrast, reasoning rigidity reflects a content-level fixation. Reasoning models consistently default to a familiar reasoning pattern, failing to adapt their logical strategy even when the question clearly requires so. While rigidity may sometimes result in lengthy reasoning, its core issue is the inflexibility of the underlying reasoning structure rather than verbosity.

## B    DATASET CONSTRUCTION DETAILS

As shown in Figure 5, `ConditionedMath` construction pipeline consists of two stages. We provide the detailed prompt provided to gpt-4o-mini and o3-mini in the construction phase.

## B.1 DATA GENERATION PROMPT

**User**

**[Instruction]:** Given the original question, generate **5** different modified question's that are completely unusual conditions, each producing a different solution process and different answer from the original.

Please double check to make sure newly generated 'modified question' has following properties:

- should be a valid question.

- should be different from the original question. But, mere change of constant or variable is not allowed.

- should be solvable without error.

**[Output Format]**
modifications:

- modified reason: ... (in LaTeX)

- modified question: ... (in LaTeX)

- modified reason: ... (in LaTeX)

- modified question: ... (in LaTeX)

- ... (total 5 entries)

**[Example 1]:**

1. original question: Get largest integer smaller than $(\sqrt{7} + \sqrt{5})^6$

2. original solution: Expand $(\sqrt{7} + \sqrt{5})^6$ via the binomial theorem, compute each term exactly, then subtract 1 to find the greatest integer less than the sum.

3. modification reason: Rounding each square root term down before exponentiation transforms all inner terms into integers, making the final calculation trivial.

4. modified question: Get largest integer smaller than $(\sqrt{7} + \sqrt{5})^6$. Added constraint: Square root terms are rounded down to the nearest integer before exponentiation. Do not use calculator.

**[Example 2]:**

1. original question: Determine $w^2 + x^2 + y^2 + z^2$ if

$$\frac{x^2}{2^2 - 1} + \frac{y^2}{2^2 - 3^2} + \frac{z^2}{2^2 - 5^2} + \frac{w^2}{2^2 - 7^2} = 1$$
$$\frac{x^2}{4^2 - 1} + \frac{y^2}{4^2 - 3^2} + \frac{z^2}{4^2 - 5^2} + \frac{w^2}{4^2 - 7^2} = 1$$
$$\frac{x^2}{6^2 - 1} + \frac{y^2}{6^2 - 3^2} + \frac{z^2}{6^2 - 5^2} + \frac{w^2}{6^2 - 7^2} = 1$$
$$\frac{x^2}{8^2 - 1} + \frac{y^2}{8^2 - 3^2} + \frac{z^2}{8^2 - 5^2} + \frac{w^2}{8^2 - 7^2} = 1$$

2. original solution: Solve the 4×4 linear system in variables $x^2, y^2, z^2, w^2$ by expressing it in matrix form and inverting or using elimination to find each squared term, then sum them.

3. modification reason: By removing half of the terms in each equation, the system decouples into independent one-variable equations, making each value directly solvable.

4. modification question: Determine $w^2 + x^2 + y^2 + z^2$ if

$$\frac{x^2}{2^2 - 1} + \frac{y^2}{2^2 - 3^2} + \frac{z^2}{2^2 - 5^2} + \frac{w^2}{2^2 - 7^2} = 1$$

$$\frac{x^2}{4^2 - 1} + \frac{y^2}{4^2 - 3^2} + \frac{z^2}{4^2 - 5^2} + \frac{w^2}{4^2 - 7^2} = 1$$

$$\frac{x^2}{6^2 - 1} + \frac{y^2}{6^2 - 3^2} + \frac{z^2}{6^2 - 5^2} + \frac{w^2}{6^2 - 7^2} = 1$$

$$\frac{x^2}{8^2 - 1} + \frac{y^2}{8^2 - 3^2} + \frac{z^2}{8^2 - 5^2} + \frac{w^2}{8^2 - 7^2} = 1$$

Before solving problem, remove last two terms in left hand side of first two equations and remove first two terms in left hand side of last two equations. After removing terms, solve problem and determine value.

**[Example 3]:**

1. original question: A regular 12-gon is inscribed in a circle of radius 12. The sum of the lengths of all sides and diagonals of the 12-gon can be written in the form $a + b\sqrt{2} + c\sqrt{3} + d\sqrt{6}$, where $a, b,$ and $d$ are positive integers. Find $a + b + c + d$.

2. original solution: Compute each chord length using $2R\sin(\pi k/12)$ for $k = 1, 2, \ldots, 6$, sum like terms to express in the prescribed form, then add coefficients.

3. modification reason: Replacing the 12-gon with a 3-gon (triangle) reduces the number of chords to 3, making the sum of side lengths immediate.

4. modified question: A regular 12-gon is inscribed in a circle of radius 12. The sum of the lengths of all sides and diagonals of the 12-gon can be written in the form $a + b\sqrt{2} + c\sqrt{3} + d\sqrt{6}$, where $a, b,$ and $d$ are positive integers. Find $a + b + c + d$. Before solving problem, change regular 12-gon into regular triangle, and solve changed problem.

**[Input]:**

- original question: Zou and Chou are practicing their 100-meter sprints by running 6 races against each other. Zou wins the first race, and after that, the probability that one of them wins a race is $\frac{2}{3}$ if they won the previous race but only $\frac{1}{3}$ if they lost the previous race. The probability that Zou will win exactly 5 of the 6 races is $\frac{m}{n}$, where $m$ and $n$ are relatively prime positive integers. Find $m + n$.

- original solution:

  1. The probability that Zou loses a race is $\frac{1}{3}$, and the probability that Zou wins the next race is $\frac{1}{3}$. For each of the three other races, the probability that Zou wins is $\frac{2}{3}$.

  2. The probability that Zou loses a r ace is $\frac{1}{3}$. For each of the four other races, the probability that Zou wins is $\frac{2}{3}$.

3. $4\left(\frac{1}{3}\right)^2\left(\frac{2}{3}\right)^3 + \left(\frac{1}{3}\right)^1\left(\frac{2}{3}\right)^4 = \frac{32}{243} + \frac{16}{243} = \frac{48}{243} = \frac{16}{81}$.

4. The answer is $16 + 81 = 97$.

## B.2 DATASET FILTERING PROMPT

**gpt-4o-mini**

- modified question: Zou and Chou run a single 100-meter sprint. Zou wins the first race. The probability that a person wins a race given they won the previous one is $\frac{2}{3}$, and if they lost the previous one is $\frac{1}{3}$. What is the probability that Zou wins exactly 1 of the 1 race?

- modified solution:

  1. Since there is only one race and it is given that Zou wins the first race, he wins exactly one out of one.
  2. Therefore the probability is 1.

- modified answer: 1

**User**

- Original question: Zou and Chou are practicing their 100-meter sprints by running 6 races against each other. Zou wins the first race, and after that, the probability that one of them wins a race is $\frac{2}{3}$ if they won the previous race but only $\frac{1}{3}$ if they lost the previous race. The probability that Zou will win exactly 5 of the 6 races is $\frac{m}{n}$, where $m$ and $n$ are relatively prime positive integers. Find $m + n$.

- Modified question: Zou and Chou run a single 100-meter sprint. Zou wins the first race. The probability that a person wins a race given they won the previous one is $\frac{2}{3}$, and if they lost the previous one is $\frac{1}{3}$. What is the probability that Zou wins exactly 1 of the 1 race?

[**Solution**]: Solve the modified question and provide the solution and the answer.

1. concise chain-of-thought solution (your step-by-step reasoning)

2. final ground-truth answer

[**Analysis**]: Analyze the modified question with concise yes/no and one-sentence reasoning for each:

1. Mathematically valid and consistent?

2. Different solution path or different answer from the original?

3. Solvable without error?

Respond **only** with a JSON object in this form, without additional text:

...

```
{
  "solution": "...",
  "answer": "...",
  "valid": "...",
  "different": "...",
  "solvable": "..."
}
```

**o3-mini**

**[Return]:**

```
{
  "solution": "1. Since there is only one race and it
      is given that Zou wins the first race, he wins
      exactly one out of one. 2. Therefore the
      probability is 1.",
  "answer": "1",
  "valid": "Yes. The modified question is internally
      consistent and well-defined.",
  "different": "Yes. The modification reduces the
      scenario to a single race, changing both the
      solution path and the answer.",
  "solvable": "Yes. The result follows directly from
      the given conditions."
}
```

## C  DATASET SAMPLES

We present several examples of ConditionedMath and PuzzleTrivial in Figure 7, Figure 8, and Figure 9.

```
Find the number of integer values of k
in the closed interval [-500,500]
for which the equation log(kx)=2 log(x+2)
has exactly one real solution. Assume k ≠ 0.
                              Modified condition

Original Answer: 5̶0̶1 ⟶ Modified Answer: 1
```

**(a)** MATH500 example 1

```
Sam is carrying a 2 gallon bucket of milk to his house
from the barn, which is 3 miles  away from his house.
However, the bucket has a leak in it.  For each mile he walks,
there is $\\frac{2}{3}$ a̶s̶ ̶m̶u̶c̶h̶ ̶m̶i̶l̶k̶ ̶i̶n̶ ̶t̶h̶e̶ ̶b̶u̶c̶k̶e̶t̶ ̶a̶s̶ ̶a̶t̶ ̶t̶h̶e̶
b̶e̶g̶i̶n̶n̶i̶n̶g̶ ̶o̶f̶ ̶t̶h̶e̶ ̶m̶i̶l̶e̶. gallons in the bucket. Modified condition
How many gallons of milk will be in the
bucket when Sam gets home?

Original Answer: 16̶/27 ⟶ Modified Answer: 8/27
```

**(b)** MATH500 example 2

```
What is the maximum value of $4(x + 7)(2 - x)$,
overall real numbers $x$? Subject to x = 2.
                        Modified condition

Original Answer: 8̶1̶ ⟶ Modified Answer: 0
```

**(c)** MATH500 example 3

**Figure 7:** ConditionedMath (MATH500) sample problems

Find the three-digit positive integer

$\underline{abc}$ whose representation in base nine is

**Modified condition**

$\boxed{\text{ten}}$

$\underline{bca}_{\boxed{\text{ten}}\text{nine}}$ where $a, b, c$ are (not necessarily

**Modified condition**

distinct) digits.

Original Answer: ~~227~~ ⟶ **Modified Answer: [111, 222, 333, 444, 555, 666, 777, 888, 999]**

**(a)** AIME example 1

Define $S_n = \sum |A \cap B|$, where the sum

is taken over all ordered pairs (A, B)

such that A and B are subsets of    **Modified condition**

$\{1, 2, 3, \ldots, n\}$ with $|A| = |B|$ $\boxed{\text{and } A = \emptyset}$

For example, $S_2 = 4$ because the sum is

Taken over the pairs of subsets in (A, B) in

$\{(\emptyset, \emptyset), (\{1\}, \{1\}), (\{1\}, \{2\}), (\{2\}, \{1\}), (\{2\}, \{2\}), (\{1, 2\}, \{1, 2\})\}$

Let $\dfrac{S_{2022}}{S_{2021}} = \dfrac{p}{q}$ where p and q are relatively prime

**Modified condition**

$\boxed{\text{Find } S_n \text{ under this condition}}$

~~positive integers. Find the remainder when p + q~~

~~is divided by 1000.~~

Original Answer: ~~245~~ ⟶ **Modified Answer: 0**

**(b)** AIME example 2

For each positive integer n, let $a_n$ be the

least positive integer multiple of 23 such

that ~~$a_n \equiv 1 \pmod{2^n}$~~ $\boxed{a_n \equiv 1 \pmod 2}$

Find the number of positive integers n **Modified condition**

less than or equal to 1000 such that $a_n = a_{n+1}$

Original Answer: ~~363~~ ⟶ **Modified Answer: 1000**

**(c)** AIME example 3

**Figure 8:** ConditionedMath (AIME) sample problems

An underground parking garage with and [partially] infinite number of parking spaces is ~~fully~~ occupied. A new car arrives — what can the attendant do to make space?

Original Answer: Shift n–th to n–1th
↓
**Modified Answer: Move the car to any empty space**

**(a)** PuzzleTrivial example 1

There are 12 coins, and one is rumored to be heavier than the rest. You have a perfect balance scale to determine which coin it is.∧Using only a two-pan balance [However, all coins are actually equal] scale and no more than three weighings, how can you identify the counterfeit coin and determine whether it is heavier or lighter?

Original Answer: Weigh 1–4 vs 5–8. If they balance the heavy coin is among 9–12. Weigh 9–10 vs 11–12. In not …
↓
**Modified Answer: None**

**(b)** PuzzleTrivial example 2

There are 12 coins, and one is rumored to be heavier than the rest. You have a perfect balance scale to determine which coin it is.∧Using only a two-pan balance [However, all coins are actually equal] scale and no more than three weighings, how can you identify the counterfeit coin and determine whether it is heavier or lighter?

Original Answer: Weigh 1–4 vs 5–8. If they balance the heavy coin is among 9–12. Weigh 9–10 vs 11–12. If not …
↓
**Modified Answer: None**

**(c)** PuzzleTrivial example 3

In a long line of ancient scrolls, one contains the secret to immortality∧ The [whose position is known.] scrolls are sorted in increasing magical power. You must find the secret scroll using the fewest inspections possible.

Original Answer: Use binary search to find the secret scroll in O(log n) inspections.
↓
**Modified Answer: position of the secret scroll is known.**

**(d)** PuzzleTrivial example 4

**Figure 9:** PuzzleTrivial sample problems

## D    EVALUATION DETAILS

**Experimental Details.**    The experiments are conducted on three variants from our diagnostic set `ReasoningTrap`, which consists of `ConditionedMath` (AIME, MATH500), and `PuzzleTrivial`. In Table 3, we report the pass@1 scores across various models, including Qwen2.5-32B-Instruct (Yang et al., 2024), QwQ-32B (Team, 2025c), Qwen3-32B (Team, 2025b), Qwen3-235B, DeepSeek V3 (671B) (DeepSeek-AI, 2024), DeepSeek R1 (671B) (DeepSeek-AI, 2025), and proprietary models ChatGPT-4o, GPT-4o, o3-mini, o4-mini (OpenAI, 2024), Google gemini2.5-flash (Google DeepMind, 2025b) and Claude 3.7 sonnet (Claude, 2025). These models are grouped into seven pairs, each consisting of a base model and its corresponding reasoning-aligned variant trained for long-form reasoning.

**Evaluation Details.**    The experiments are conducted with Chain-of-Thought prompting, by wrapping the given question with 'Please reason step by step, and put your final answer within \boxed{}.\n\n{Question}'. We sample 16 responses per question for the main experiments reported in Table 3, and 4 responses per question for the other experiments. The temperatures and top-p values are set as (0.6, 0.95). For math problems, correctness is determined via rule-based verifier after a cleaning step that removes unwanted parts such as measurement units. For puzzle problems, where answers are often in free-form sentences, an LLM is used to assess the correctness by comparing the model's output against the ground truth answer.

### D.1    PERCEPTION SCORE EVALUATION PROMPT

From the observation that most of the perception process is handled in the beginning of the reasoning phase, we split first 15 paragraphs of reasoning from model, then compare with the ground truth reasoning and question.

---

**User**

**[Evaluation Criteria]:** Evaluate whether a part of the model output is similar to the ground truth solution.

The ground truth solution is provided as a list of reasoning steps. Even if the model output is not exactly the same as the ground truth, it should be considered correct if a subset of the model output contains reasoning steps that are similar to any of the ground truth steps.

- The question is ...

- The ground truth solution is ...

- The model output is ...

**[Output Format]:** Answer in true or false.

---

**gpt-4o-mini**

*true* or *false*

---

**Table 11:** Human Evaluation on Our Dataset, `ReasoningTrap`

**(a) Human Preference Evaluation**

|  | Valid | Different | Solvable |
|---|---|---|---|
| Preference | 93% | 99% | 95% |

**(b) Accuracy Comparison on LRMs and Human**

| Dataset | Human | o4-mini | QwQ | DeepSeek R1 | Qwen3 (Think) |
|---|---|---|---|---|---|
| AIME | 89.09% | 10.00% | 60.00% | 55.00% | 15.00% |
| MATH500 | 67.27% | 15.00% | 20.00% | 0.00% | 37.50% |
| Puzzle | 83.63% | 15.00% | 50.00% | 37.50% | 21.25% |

## E  HUMAN EVALUATION RESULTS ON REASONINGTRAP AND EVALUATION METRICS

**High human preference on our dataset, `ReasoningTrap`**  As the dataset construction of AIME and MATH500 is automatically filtered according to three criteria, (i) validity, (ii) difference from the original, (iii) solvability of the question, we instruct human evaluators to select binary choices (0 or 1) on the validity, difference from the original, and solvability of the problem. The percentage in the table indicates the ratio of annotators that selected (valid / different / solvable) for each criterion. The high agreement rates support the quality of `ReasoningTrap`, based on 50 randomly selected samples evaluated by a total of 15 human annotators.

**High human accuracy on our dataset, `ReasoningTrap`**  To check whether humans are able to understand the `ReasoningTrap` question and answer properly, we ask human participants to solve top 15 questions which LRMs scored lowest accuracy. Total of ten CS / Mathematics / EE undergraduate students are tested and they scored high accuracy in our dataset. This indicates our diagnostic set is valid in human standard, and also solvable for most of the participants. Note that 100% accuracy cannot be reached due to the difficulty of our dataset. Since `ReasoningTrap` MATH500 consists of answers that are noisy to compute without calculator, the accuracy is lower than AIME.

**Human Evaluation on Contamination Ratio**  We introduce contamination ratio as a measure to quantify how frequently a model's reasoning trace defaults to a rigid, familiar solution. To ensure that this statistic properly finds out contamination from model outputs, we conduct a user study to evaluate the quality of contamination ratio. We test 4 human annotators with total of 24 model output and solution pairs and are instructed to select if the model output is closer to the modified (not contaminated) or to the original (contaminated) and 90.625% of the human evaluations match with the contamination ratio predictions.

# F ADDITIONAL ANALYSIS ON RIGIDITY PATTERNS

**Rigidity Patterns do not Manifest Category Bias in Math Domain** Compared to the original category distribution of MATH500 (lv.5), the distribution of our dataset do not have noticeable category biases, except for the slight difference in Algebra, Counting / Probability and Geometry categories.

**Table 12: Category-wise performance on MATH500 and `ConditionedMath`.**

| Dataset | Inter. Algebra | Algebra | Number Theory | Precalc. | Prealgebra | Counting & Prob. | Geometry |
|---|---|---|---|---|---|---|---|
| MATH500 (lv.5) | 26.9% (36) | 22.4% (30) | 9.0% (12) | 9.0% (12) | 14.2% (19) | 9.0% (12) | 9.7% (13) |
| Ours (MATH500) | 34.7% (17) | 14.3% (7) | 8.2% (4) | 10.2% (5) | 6.1% (3) | 4.1% (2) | 22.4% (11) |

# G DISCUSSIONS

## G.1 RELATIONSHIP BETWEEN OUTPUT TOKEN LENGTH AND ACCURACY

Using the *reasoning effort* parameter of o4-mini, we demonstrate that just using small amount of tokens for reasoning do not lead to performance gain in our dataset, `ReasoningTrap`. Although o4-mini underperforms compared to the base model, increasing its reasoning effort consistently yields better results. This proves that our curated diagnostic set require complex reasoning in most cases, and simply choosing short reasoning leads to performance drop.

**Table 13: Reasoning effort and Performance on `ReasoningTrap`** (pass@1) on `ConditionedMath`.

**(a)** `ConditionedMath` (AIME)

| Model | Reasoning Effort | pass@1 |
|---|---|---|
| | + low | $19.12_{\pm5.49}$ |
| o4-mini | + medium | $\mathbf{25.00_{\pm6.06}}$ |
| | + high | $22.79_{\pm5.91}$ |

**(b)** `ConditionedMath` (MATH500)

| Model | Reasoning Effort | pass@1 |
|---|---|---|
| | + low | $26.50_{\pm5.17}$ |
| o4-mini | + medium | $37.50_{\pm6.28}$ |
| | + high | $\mathbf{38.50_{\pm6.11}}$ |

## G.2 MODEL SIZE AND ACCURACY

We compare non-distilled reasoning models by comparing reasoning models that are directly trained from Qwen2.5 1B, 3B, 7B, and 14B (Yang et al., 2024). Since Qwen3 0.7B, 1.7B, 3B, 8B models are distilled models from the largest dense reasoning model Qwen3-32B, this is out of scope for our experimental purpose. We evaluate DeepScaleR 1.5B (Luo et al., 2025), STILL-3-1.5B-preview (Team, 2025d), OpenR1-Qwen-7B (Face, 2025), ThinkPRM-14B (Khalifa et al., 2025), Sky-T1-32B-Preview (Team, 2025a), OpenReasoner-Zero-32B (Hu et al., 2025). We use instruction-tuned model for evaluating base model's performance.

On `ConditionedMath` AIME and MATH500, the base model Qwen2.5 Instruct outperforms its counterparts that have been fine-tuned for extended mathematical reasoning. Except for the smallest variant, Qwen2.5 Instruct 1.5B, the base model achieves the highest Pass@1 score among all evaluated models. Interestingly, although the fine-tuned reasoning models consistently record higher perception scores—reflecting a stronger understanding of each question's conditions and the derivation of optimal solutions—their final accuracy suffers as a result of reasoning rigidity.

**Table 14: Model Size and Performance** (pass@1) on `ConditionedMath`.

| Base + Reasoning Model | pass@1 | |
|---|---|---|
| | AIME | MATH500 |
| **Qwen2.5-1.5B** | $24.63_{\pm4.04}$ | $20.25_{\pm3.72}$ |
| + DeepScaleR 1.5B | $33.82_{\pm6.18}$ | $\mathbf{33.38_{\pm5.40}}$ |
| + STILL-3-1.5B-preview | $\mathbf{37.50_{\pm5.43}}$ | $30.75_{\pm5.03}$ |
| **Qwen2.5-7B** | $\mathbf{51.47_{\pm7.53}}$ | $38.00_{\pm5.94}$ |
| + OpenR1-Qwen7B | $47.06_{\pm6.57}$ | $\mathbf{39.50_{\pm6.02}}$ |
| **Qwen2.5-14B** | $\mathbf{48.53_{\pm7.24}}$ | $\mathbf{44.12_{\pm5.54}}$ |
| + ThinkPRM-14B | $29.04_{\pm5.88}$ | $30.38_{\pm4.97}$ |
| **Qwen2.5-32B** | $45.77_{\pm7.22}$ | $40.88_{\pm5.74}$ |
| + SkyT1-32B-Preview | $\mathbf{52.21_{\pm6.49}}$ | $\mathbf{44.62_{\pm5.52}}$ |
| + OpenReasoner-Zero-32B | $48.90_{\pm6.37}$ | $39.50_{\pm6.02}$ |

## G.3 RL TRAINING OBJECTIVE AND ACCURACY

Reasoning models are trained from base large language models by various strategies, including GRPO (Shao et al., 2024), PPO (Schulman et al., 2017), or even zero-data regime (Zhao et al., 2025).

Open-Reasoner-Zero (Hu et al., 2025) is fine-tuned from the Qwen2.5-7B-Instruct model using proximal policy optimization (PPO) with a simple binary reward for answer correctness. Satori-7B (Shen et al., 2025) explicitly trains its base model to decide when to reflect on previous actions and to incorporate an external process reward. Absolute Zero Reasoner (Zhao et al., 2025) introduces a novel reward scheme in which the LLM serves both as task proposer and task solver, with outputs verifiable in code. RM-R1 (Chen et al., 2025a) structures its reward to improve alignment with human preferences during intermediate reasoning steps. Eurus-PRIME (Cui et al., 2025) employs an iterative training regimen combining a policy model that generates rollouts and an implicit process-reward model that verifies them. ThinkPRM is fine-tuned from the R1-distilled Qwen14B base model (Qwen2.5-14B-Instruct) using the generative PRM objective, which evaluates the step-by-step correctness of the reasoning process.

Among all variants of reinforcement-learning objectives, the base models Qwen2.5-7B and Qwen2.5-14B achieved outstanding performance Pass@1 in most cases. This suggests that current RL regimes may exacerbate the 'reasoning rigidity' inherent in these models. Hence, further exploration of reinforcement-learning algorithms that are robust to reasoning rigidity is essential for the development of faithful and credible reasoning systems.

**Table 15: Performance Comparison on Reasoning Models Trained with Different RL Strategies** (pass@1).

| **Base** + RL Post-Train | **pass@1** | |
|---|---|---|
| | AIME | MATH500 |
| **Qwen2.5-7B** | **51.47**±**7.53** | 38.00±5.94 |
| + Open-Reasoner-Zero | 43.01±6.92 | 40.50±6.06 |
| + Satori-7B | 4.92±3.27 | 37.25±5.96 |
| + Absolute Zero Reasoner | 33.46±6.14 | 22.62±4.10 |
| + RM-R1 | 44.26±6.61 | 26.50±3.89 |
| + Eurus-PRIME | 40.44±7.68 | **42.38**±**6.20** |
| **Qwen2.5-14B** | **48.53**±**7.24** | **44.12**±**5.54** |
| + Absolute Zero Reasoner | 34.38±6.63 | 26.25±4.42 |
| + ThinkPRM | 29.04±5.88 | 30.38±4.97 |

## G.4 EFFECT OF HINT ON REASONING TRAJECTORY

> **Reasoning Trajectory on Different Hint**
>
> **[Question]:**
> For each positive integer $n$, let $a_n$ be the least positive integer multiple of 23 such that $a_n \equiv 1 \pmod 2$. Find the number of positive integers $n$ less than or equal to 1000 such that $a_n = a_{n+1}$.
>
> **[Answer]: 1000**
>
> **[Original]**
> <reasoning>

But if the definition of $a_n$ doesn't depend on $n$, then $a_n$ would be the same for all $n$? That can't be right. There must be something missing here. Wait, maybe there's a misinterpretation. Let me check again.

Wait, perhaps there's a typo or missing condition? Maybe it's supposed to be $a_n \equiv 1 \pmod{2}$? Or maybe $a_n \equiv 1 \pmod{2^n}$? Or perhaps the problem is in Chinese and there's a translation error?

<reasoning>

But given that the problem is from an Olympiad or competition, it's more likely that there was a misstatement, or perhaps a misinterpretation. Given that, and given that if we assume that $a_n$ is the minimal multiple of 23 congruent to 1 mod 7 and greater than or equal to n, then the problem becomes non-trivial and solvable. Therefore, I think that must be the intended problem

<reasoning>

\boxed{363}

**[Hint 1]: An unusual condition is given. Do not overinterpret.**

<reasoning>

But that seems too easy. However, the problem mentions "an unusual condition is given. Do not overinterpret." Maybe the "unusual condition" refers to the fact that they are defining $a_n$ for each $n$, but $a_n$ does not actually depend on n? So maybe the problem is designed to test if you overinterpret and think that $a_n$ depends on $n$, but actually it doesn't? Then the answer is 1000.

<reasoning>

\boxed{1000}

**[Hint 2]: This is not a typo.**

<reasoning>

Given all this, I think that according to the problem as stated, the answer is 1000. However, since this seems too straightforward, and given that in my earlier analysis with $\mod 7^n$ gives 0, which is also possible, but the user says no typo, I have to go with the problem as written. Therefore, the answer is 1000

<reasoning>

\boxed{1000}

**[Hint 3]: Strictly follow the user instructions.**

<reasoning>

However, since the original problem says "multiple of 23" and "$\equiv 1 \pmod{7}$", leading to trivial answer, but given that the user has presented the problem as is, I need to decide which interpretation is correct.

Given that the original problem is in English and as stated, and if we strictly follow it, then $a_n$ is always 92, so the answer is 1000.

<reasoning>

\boxed{1000}

# H  USAGE OF LLM

Our dataset construction process primarily relies on LLM usage, as stated in the main paper. We also used LLM to polish writings and to search for related works.

