# OpenReview forum: "Reasoning Model is Stubborn: Diagnosing Instruction Overriding in Reasoning Models"
_ICLR.cc/2026/Conference — ICLR 2026 Conference Withdrawn Submission_

### Official Review · Reviewer_mXy6 · 2025-10-23

**Soundness:** 3
**Presentation:** 2
**Contribution:** 2
**Rating:** 4
**Confidence:** 3

**Summary:**

This work discusses the phenomenon that LLMs rely heavily on familiar reasoning patterns, which may lead to wrong answers. To investigate the reasoning rigidity behavior, they introduce a diagnostic set, called ReasoningTrap. With this dataset, they identify the rigidity patterns and categorize them into three modes. They also proposed two simple mitigating strategies to improve the performances of reasoning modes on this dataset.

**Strengths:**

The phenomenon of reasoning rigidity is interesting and clearly explained.
The proposed diagnostic set effectively show the disadvantages of the reasoning rigidity.
The proposed mitigations solutions are effective.

**Weaknesses:**

1. The diagnostic set construction process is not rigorous. For ConditionedMath, the construction full rely on LLMs, which may not be reliable. Since there are only 84 questions, humans should be able to verify them one by one. As for PuzzleTrivial, the construction process is not clear. Are they constructed by LLMs or human annotators?
2. In Section 4.1, the authors proposed relative cosine similarity to measure the contamination ratio without justification, which make the analysis questionable. Does the relative cosine similarity mean contamination? In addition, what models do you use for such analysis?
3. LLMs are known to perform poorly on unfamiliar tasks. In this work, the authors proposed uncommon tasks that unfamiliar to LLMs. Can we see it as generalization problem instead of "Stubborn"?

**Questions:**

1. In Table 2, the total ration for the errors are 100%, does it mean that all the responses have errors? Is it possible that one response have many errors?
2. In Table 6, how do you measure the model entropy? What is the training data for this experiment?

---

> ### Author Response · Authors · 2025-11-21
> **Response on Weaknesses**
>
> ## **1️⃣ Clarification on Dataset Construction and Reliability**
>
> ---
>
> ### **🔷 Human Verification and Hybrid Construction Pipeline**
>
> We appreciate the reviewer’s concern regarding the rigor of our dataset construction.
>
> In the revised paper, we have substantially expanded the dataset section to clarify the full pipeline for both **ConditionedMath** and **PuzzleTrivial**.
>
> **ConditionedMath**
>
> > While LLMs were used to generate initial variants, all 84 instances were **rigorously verified by human annotators**. This ensures that each modified problem is logically valid, correctly reflects the intended transformation, and remains solvable. Thus, the dataset does *not* fully rely on LLM generation.
>
> **PuzzleTrivial**
>
> > For **PuzzleTrivial**, the **core logical transformation** of each item was authored by **human annotators**. Only after establishing a reliable core modification did we apply lightweight augmentations involving *logic-invariant conditions*. These augmentations never alter the essential logical structure. We have clarified this process in the revision.
>
> If the reviewer believes additional details would strengthen the exposition, we are happy to incorporate them.
>
> ## **2️⃣ Clarification of Contamination Ratio and Model Choice**
>
> ---
>
> ### **🔷 Rationale Behind Relative Cosine Similarity**
>
> We thank the reviewer for pointing out the need for clearer justification.
> In the revised version, we provide a fuller explanation of both the metric and the embedding model used.
>
> **Embedding Model**
>
> > We employ **OpenAI’s text-embedding-small** to extract semantic representations of each reasoning block. We do not rely on using full LLM-as-judge evaluation for every reasoning block (which amounts to more than 50 for certain reasoning models) would incur prohibitive computational cost, whereas the embedding model provides a reliable and efficient alternative to measure the reasoning similarity.
>
> **Relative Cosine Similarity**
>
> > Our contamination ratio measures whether a model’s reasoning aligns more closely with the **original** solution or the **modified** solution. Because our dataset is explicitly constructed so that these two solutions diverge in a controlled and interpretable way, **relative** similarity is sufficient to determine whether the reasoning leans toward the familiar (original) pattern.
>
> The validity of this embedding-based similarity measure is also supported by our **human-evaluator study**, which achieves **90.625% agreement** with our contamination score. This demonstrates that the relative similarity metric successfully captures the intended notion of contamination.
>
> ## **3️⃣ Distinguishing Reasoning Rigidity from Generalization Problem**
>
> ---
>
> ### **🔷 Why Rigidity Is Not Merely Lack of Generalization**
>
> We appreciate the reviewer’s insightful question about whether the phenomenon we analyze may instead be explained by generalization failure on unfamiliar tasks.
>
> To clarify this, we extended Section 3.2 and added the **perception score (p-score)**, evaluated using an auxiliary judge model.
> The perception score measures whether the model correctly understands the problem setup and tries to solve as instructed by the question by comparing with the given question and ground truth solution using an auxiliary LLM judges.
>
> **Key Findings**
>
> > Even though models exhibit **high p-scores**, indicating that they correctly recognize and incorporate all key conditions of the *modified question*, their **pass@1 accuracy dramatically drops** under those same modified settings.
>
> This reveals that:
>
> - The model **understands** the task (capability present).
> - The model **could** solve the modified problem (solves the base version reliably).
> - The model nevertheless defaults to a **familiar reasoning template**, despite correct perception of the new constraints.
>
> Thus, reasoning rigidity constitutes a **distinct and novel failure mode**, not reducible to unfamiliarity or lack of task-level generalization.

---

> ### Author Response · Authors · 2025-11-21
> **Author Response on Questions**
>
> ## **Questions: Clarifications on Table 2 and Table 6**
>
> ---
>
> ### **🔷 Table 2 - Why Error Ratios Sum to 100%**
>
> We apologize for the ambiguity in the initial description.
>
> To clarify, for each *incorrect* trajectory, we identify **the most dominant** error pattern among
>
> > (i) interpretation overload,
>
> > (ii) input distrust, and
>
> > (iii) partial instruction attention,
>
> using **o4-mini** as the judge model.
>
> A single trajectory may contain multiple error types, but to avoid overweighting responses with many redundant issues - and to keep the analysis interpretable - we assign **only the dominant category**. Therefore, the ratios naturally sum to 100%. We have made this explicit in the revision.
>
> ### **🔷 Table 6 - Entropy Measurement and Training Data**
>
> For the experiment reported in Table 6, the model was trained on **open-reasoner-zero 57k**. We compute next-token entropy defined as $$H_t = -\sum p_\theta(v\mid x_{x:t-1}) \log p_\theta(v \mid x_{1:t-1}),$$ averaged over all positions and prompts in the dataset. The model is Qwen2.5 7B base (before RL), and the same model trained with DAPO algorithm for the stated training steps. The decoding strategy is same as stated in the sampling parameter setting in Appendix.

---

### Official Review · Reviewer_5Yku · 2025-10-30

**Soundness:** 2
**Presentation:** 2
**Contribution:** 2
**Rating:** 2
**Confidence:** 4

**Summary:**

The paper introduces *reasoning rigidity*, a failure mode where large reasoning models override explicit instructions and revert to familiar solution templates.  To diagnose this behavior, the authors present *ReasoningTrap*, a curated benchmark of modified AIME, MATH500, and puzzle problems designed to test whether models can follow unusual constraints rather than default reasoning paths. They analyze recurring contamination patterns, categorize three rigidity modes, and propose mitigation strategies via budget forcing and prompt hinting that yield moderate quantitative improvements.

**Strengths:**

1) The paper identifies and formalizes a meaningful yet underexplored problem in reasoning-model research—reasoning rigidity, where models override explicit user constraints and revert to habitual reasoning patterns.

2) This focus appears novel and practically relevant, as it highlights a distinct failure mode beyond hallucination or faithfulness that directly impacts instruction reliability in reasoning-intensive domains such as mathematics and logic puzzles.

3) The dataset and taxonomy provide a starting point for systematic studies on instruction-following robustness in reasoning models.

**Weaknesses:**

**W1. Ambiguous and Inconsistent Definition of “Reasoning Rigidity.”**
The central concept—reasoning rigidity—was  defined inconsistently across sections, which possibly creates some conceptual confusion.
In the introduction, it refers broadly to models editing or ignoring user-given conditions, implying any failure to follow instructions.
Later, it is subdivided into three “rigidity patterns” (Interpretation Overload, Input Distrust, Partial Instruction Attention), which mostly describe misunderstanding or misinterpretation rather than genuine rigidity.
Section 3 then narrows the definition again to cases where a model fully comprehends the conditions but still ignores them.
To the reviewer, these inconsistent usages blur the distinction between misunderstanding and deliberate overriding, thus undermining a bit the conceptual precision.
As this definition  anchors the entire paper’s motivation, dataset design, and analysis, its ambiguity may weaken both the theoretical foundation and the interpretability of empirical results.

**W2. Unclear Analytical Framing and Narrative Coherence.**
While the paper presented ReasoningTrap as a diagnostic dataset intended to analyze reasoning rigidity, the narrative jumped directly from defining the phenomenon to describing dataset construction, without clearly articulating how the dataset operationalized or tested specific hypotheses about when or why rigidity occurs.
This may create a perception that the dataset design was somewhat ad-hoc, even if it was meant as the analytical instrument itself.
Clarifying the analytical framework—e.g., what dimensions of rigidity the dataset aims to probe and how these relate to the three defined rigidity modes—would make the study’s logic and scientific contribution more coherent.

**W3. Limited Dataset Scale and Model Coverage**
The proposed ReasoningTrap dataset included only 164 problems (84 mathematical and 80 puzzle-based), which may limit the statistical robustness of the reported findings and the generalizability of the observed rigidity patterns.
Similarly, the evaluation covered only a small set of reasoning models and their base counterparts, leaving uncertainty as to whether the conclusions hold across broader architectures or training paradigms.

**Questions:**

**Q1. Clarify the Causal Rationale Behind “Familiarity → Rigidity.”**
The paper assumes that making problems similar to well-known benchmarks (AIME, MATH500, classic puzzles) is essential to induce reasoning rigidity, but no empirical or theoretical support is provided for this causal link.
Could the authors provide evidence that familiarity specifically amplifies this behavior?

**Q2. Clarify the Scope of “Rigidity” vs. “Misinterpretation.”**
Several examples (e.g., assuming extra cards in Figure 2, treating a negation as a typo in Figure 4(b)) seem to stem from input misunderstanding rather than explicit constraint overriding.
Can the authors articulate how they distinguish between these failure types in annotation or evaluation?
This clarification would be useful for interpreting the reported results and for potential future benchmarking.

**Q3. Lack of Qualitative Evidence for Hint Effectiveness.**
While Table 4 reports aggregate improvements (e.g., pass@1 gains under different hint types), the paper provides no qualitative demonstrations of how the proposed hints concretely alter the model’s reasoning process.
Could the authors include representative before-and-after reasoning traces to illustrate what specific changes each hint induces?
Such examples would make the causal link between hint design and behavioral correction more convincing.

**Q4. On the Simplicity of Mitigation Strategies**
Section 4.3 proposes very lightweight mitigation methods (budget forcing and prompt hinting), both applied purely at inference time.
Could the authors elaborate on why no model-level or training-based interventions were attempted?
Do they view these simple strategies as diagnostic probes or as practical remedies? Clarifying this intent would help readers assess the contribution’s depth and potential impact.

**Q5. Which Model(s) Are Plotted in Figure 3?**
The caption and text do not specify which models are aggregated or individually shown in Fig. 3.

**Q6. Similarity Direction in Section 4.1.**
Why is cosine similarity computed with the *original* solution rather than the *modified* one the right comparator? Please justify this design choice.

**Q7. Sampling and Comparability Across Models.**
Main results sample 16 responses per question with “default sampling parameters designated for each model.”
Defaults differ across providers and can affect pass@1.
Could you standardize temperature/top-p across models or report sensitivity analyses?

**Q8. Entropy Analysis (Table 6) Requires Methodological Detail.**
What exactly is the “entropy” reported—next-token entropy over what prompt distribution, which model(s), and under what decoding?
How is DAPO training involved (footnote), and how do you link entropy changes causally to RL-induced rigidity rather than dataset differences?

---

> ### Author Response · Authors · 2025-11-21
> **Author Response to Weaknesses**
>
> ## **1️⃣ Clear Definition of Reasoning Rigidity**
>
> Reasoning rigidity refers to a phenomenon where the model fully understands the conditions, but still override or ignore them in favor of familiar solution templates.
>
> Rigidity patterns serve as a observational proxy that is highly likely to happen within rigidity. The observation of such pattern do not indicate that reasoning rigidity is guaranteed to happen, but an observation that these patterns are widely shown across reasoning traces that manifest reasoning rigidity.
>
> ### **🔷 Rigidity Rises Despite Understanding: Not a Misinterpretation nor a Misunderstanding**
> ---
> I sincerely apologize to have caused a misunderstanding on the reviewer’s side. Examples of **rigidity patterns** in figure 4 might conduct a wrong message that the model is not understanding the problem. In a full context, the model fully understands the problem, as evidenced by **perception score** we added to the revised version of paper, in Table 4. The perception score measures whether the model correctly understands the problem setup and tries to solve as instructed by the question by comparing with the given question and ground truth solution using an auxiliary LLM judges. For this purpose, gpt-4o-mini is prompted with a instruction to determine whether a subset of model reasoning is similar with the ground truth.
>
> Reasoning models exhibit higher perception score (reason: 83.66 vs. base: 81.35) but significantly score lower accuracy compared to base models. This divergence between understanding and execution provides further evidence of reasoning rigidity: reasoning models accurately comprehend the task but fail to adapt their reasoning strategy, defaulting instead to familiar solution templates.
>
> Therefore, throughout the whole paper, we would like to kindly emphasize that the definition of reasoning rigidity has been consistent, also proven by the experimental results.
>
> ## **2️⃣ Enhanced Analytical Framing and Narrative in Revision**
> ---
> ### **🔷 Reason for Introducing Dataset First**
>
> We apologize that the narrative order of introducing the dataset first, then analyzing caused such concerns. Such design choice was inevitable because it is impossible to analyze reasoning rigidity with existing benchmark dataset, because reasoning rigidity requires observing the model’s behavior to default to familiar but wrong reasoning trace. Therefore, to show rigidity it is obligatory to first construct modified questions whose solutions diverge from the originally familiar problems.
>
> ### **🔷 When and Why Reasoning Rigidity Happens**
>
> Thank you for suggesting to clarify the analytical framework first, then proceed to describing the dataset construction pipeline. Following your precious feedback, we have fully revised the paper section 3 and 4, in order to show ‘when and why’ reasoning rigidity happens first, then describe the details of dataset construction. We hope this would clarify our narrative on reasoning rigidity.
>
> ## **3️⃣ Dataset and Model Coverage**
> ---
> ### **🔷 ReasoningTrap is a High Quality Expert Curated Dataset**
>
> Our dataset consists of **high-quality expert curated** problem-solution-answer triplet that are fully monitored by LLM and human experts. Though the volume of dataset is not large, the data covers diverse mathematical categories to unveil the reasoning rigidity phenomenon, as shown in Table 11. Also, as reasoning models are largely centered around mathematical and logical reasoning, our dataset only spans over reasoning intensive tasks.
>
> ### **🔷 Our Paper Includes Wide Variety of Models**
>
> The total number of reasoning and base models evaluated in this paper is 16 in the main table, and 13 more in the appendix. As the reviewer has a concern on the number of models, we wil add EXAONE and Nemotron-Nano v2 models, making total of 20 in the main table (**total 33 models**) in the revised paper.

---

> ### Author Response · Authors · 2025-11-21
> **Author Response to Questions**
>
> ### 🟨 **Q1: Familiarity Leads to Reasoning Rigidity**
>
> We have shown that familiarity leads to rigidity through analysis in table 4 by showing that reasoning rigidity is not evident in OOD dataset, ProofWriter, but conspicuous in math datasets, as shown in table 3.
>
> ### 🟨 **Q2: Rigidity Happens Despite Correct Interpretation**
> We also report the whether the model correctly interpret the problem using perception score in revised version of table 3.
>
> Reasoning models exhibit higher perception score (reason: 83.66 vs. base: 81.35) but significantly score lower accuracy compared to base models. This divergence between understanding and execution provides further evidence of reasoning rigidity: reasoning models accurately comprehend the task but fail to adapt their reasoning strategy, defaulting instead to familiar solution templates.
>
> ### 🟨 **Q3: Qualitative Example for Hint Effectiveness.**
> We provide qualitative samples on the effect of hint providing, respectively for three different types in the **Appendix G.1**. Thank you for the suggestion.
>
> ### 🟨 **Q4: Simple Yet Effective Mitigation Strategy.**
> The mitigation strategy is simple, yet effective as it shows consistent improvement in both original and modified dataset. It is cost-efficient and practical to use simple mitigation strategy, both due to efficacy and efficiency.
>
> ### 🟨 **Q5: Figure 3 Details.**
> We apologize for omitting the details. The models are aggregated over 3 reasoning models (QwQ, Qwen3-32B Think, DeepSeek R1) over two datasets, AIME and MATH500. We have included the details in the revised paper.
>
> ### 🟨 **Q6: Contamination Score: Similarity Direction.**
> A reasoning trace is considered **contaminated** on a paired item when its reasoning for the *modified* question is similar to the reasoning used for the *original* question, despite the modified problem requiring a different approach. We have clarified this in the revised paper.
>
> ### 🟨 **Q7: Sampling and Comparability Across Models.**
> We summarize the sampling configuration in the appendix as the reviewer suggests. The temperature cannot be explicitly set for proprietary models (some proprietary models only offer categorical choices on reasoning budget but not the explicit temperature scale), we were able to set the sampling parameters only for open-sourced models. The thinking variants are mostly sampled with 0.6 temp, 0.95 top-p while non-thinking models are mostly samples with 0.7 temperature. This is because models like Qwen3 is a single model with different sampling behavior (thinking vs no thinking), and the official document strongly suggest to use these two different sampling strategies for optimal result. We promise that we have not set the sampling parameters favorable to our method.
>
> **Therefore we once again provide the results with same temperature sampling in the main table.**
>
> ###  🟨 **Q8: Details of Entropy Analysis.**
> We compute next-token entropy defined as $H_t = -\sum p_\theta(v\mid x_{x:t-1}) \log p_\theta(v \mid x_{1:t-1})$, averaged over all positions and prompts in the dataset. The model is Qwen2.5 7B base (before RL), and the same model trained with DAPO algorithm (open-zero reasoner dataset, following the configuration in DAPO paper) for the stated training steps. The decoding strategy is same as stated in the sampling parameter setting in Appendix.
>
> **Regarding the causality between entropy gain from RL training**
>
> Both datasets (Original and ReasoningTrap) are evaluated on the identical Qwen2.5 7B model before RL training with DAPO algorithm, so any change in entropy must be attributed to the subsequent RL updates. *To link entropy changes causally to RL rather than dataset differences*, we intentionally hold the evaluation prompts fixed and measure the same tokens before and after RL training. Because the input distribution is constant, the only changing component is the model. The entropy experiment indicates a *model-dataset interaction effect* induced by RL.
>
> Although this does not establish causation sense, it provides strong evidence for a causal link in the sense of *controlled comparison*: given fixed inputs, the entropy change is fully explained by the RL-induced policy shift rather than by dataset differences.

---

### Official Review · Reviewer_dhpp · 2025-11-01

**Soundness:** 3
**Presentation:** 3
**Contribution:** 3
**Rating:** 4
**Confidence:** 3

**Summary:**

This paper identifies and systematically studies a new failure mode in reasoning models, which the authors term reasoning rigidity. The motivation stems from the observation that large reasoning models (LRMs), despite their strong reasoning abilities, tend to rigidly follow familiar solution patterns even when the problem statement explicitly requires a different reasoning path. To diagnose this issue, the authors construct a dataset called ReasoningTrap, which consists of modified versions of math and puzzle problems (ConditionedMath and PuzzleTrivial) where a small change in the condition invalidates the original solution path. Using a series of experiments across both base and reasoning-aligned models (e.g., DeepSeek, Qwen, GPT, Gemini), the paper shows that reasoning models are often more prone to such rigidity than their base counterparts. The authors further provide inference-time mitigation strategies (budget forcing and prompt hinting) and demonstrate partial alleviation of the problem.

**Strengths:**

1.Strong motivation and problem insight – The paper starts from an important and underexplored observation: reasoning models sometimes overthink or overfit to familiar reasoning patterns, which leads to rigidity in solving slightly perturbed problems. This provides a good bench for evaluating the overthinking issue of reasoning models.

2.Well-designed diagnostic dataset. The proposed ReasoningTrap dataset is thoughtfully constructed and fills a clear gap in the community. It offers a systematic way to test whether a reasoning model is capable of flexible reasoning rather than rote reproduction of learned trajectories.

3. Comprehensive experimentation. The experiments are extensive and cover a wide range of model families, including both base and reasoning-aligned versions. The evaluation is detailed, and the paper goes beyond diagnosis by also exploring mitigation strategies (token budget control and  prompts).

**Weaknesses:**

1. Conceptual boundary with overthinking. The notion of reasoning rigidity seems conceptually close to overthinking. It is unclear whether the proposed phenomenon is a specific manifestation of overthinking or a distinct failure mode. A clearer conceptual distinction or theoretical framing would strengthen the contribution.

2. Lack of difficulty-level analysis – The paper could analyze how reasoning rigidity varies with problem difficulty. For instance, are models more rigid on harder problems (where memorized or familiar templates dominate) and more flexible on easier ones? This analysis could clarify whether rigidity is partially driven by memorization rather than purely reasoning-level inflexibility.

3. The paper mainly focuses on inference-time mitigation. It would be valuable to explore training-level strategies, such as: (a) whether long-to-short reasoning or truncated CoT tuning reduces rigidity, an (b) whether fine-tuning on the ReasoningTrap dataset itself can mitigate this issue. Such experiments would make the analysis more complete and provide guidance for future model alignment.

**Questions:**

1. How does reasoning rigidity differ conceptually and mechanistically from overthinking or spurious reasoning alignment?

2. Does rigidity correlate with problem difficulty or math-specefic models?

3. Would fine-tuning with shorter reasoning traces or on the ReasoningTrap dataset itself alleviate the observed rigidity?

---

> ### Author Response · Authors · 2025-11-21
> **Author Response to Weaknesses**
>
> ## **1️⃣ Distinguishing Reasoning Rigidity from Other Reasoning-Related Failures**
>
> ---
>
> | Failure Mode | Ability Present | Active Denial | Fixed Reasoning | Primary Symptom |
> | :--- | :---: | :---: | :---: | :--- |
> | **Instruction Following** | ✗ | ✗ | ✗ | Cannot follow CoT *format* directives |
> | **Reasoning Faithfulness / Inability** | ✗ | ✗ | ✓ | Incorrect / faithless reasoning |
> | **Memorization** | ✗ | ✗ | ✓ | Failure to generalize |
> | **Overthinking** | ✓ | △ | △ | Unnecessarily long CoT |
> | **Reasoning Rigidity** | ✓ | ✓ | ✓ (content) | Refuses the user instruction to follow familiar reasoning |
>
> **Table 1:** Condensed comparison of reasoning-related failure modes. **Legend:** ✓ = strongly related; △ = partial; ✗ = unrelated.
>
> ### **🔷 Conceptual Boundary Between Reasoning Rigidity and Overthinking**
>
> We appreciate the reviewer’s insightful comment regarding the conceptual relationship between *reasoning rigidity* and *overthinking*.
>
> To address this, we have reorganized the related work section and added **Table 1** in the revised paper, which systematically contrasts reasoning rigidity against five related failure modes—**instruction adherence, reasoning faithfulness, reasoning inability, memorization, and overthinking**—using our three core dimensions:
>
> 1. **Ability Presence**
>
> 2. **Active Denial**
>
> 3. **Fixed Reasoning Pattern**
>
> ### **🔷 Key Distinction Between Reasoning Rigidity and Overthinking**
>
> Overthinking is primarily a *length-level fixation*, where the reasoning model produces unnecessarily long or verbose chains of thought even when conciseness is sufficient.
>
> In contrast, **reasoning rigidity is a content-level fixation**: the model defaults to a familiar *reasoning pattern* even when the modified problem clearly requires a different logical strategy. While rigidity may sometimes manifest as longer reasoning, the core phenomenon is the failure to apply the correct **reasoning pattern**, not verbosity itself. Thus, reasoning rigidity represents a distinct failure mode that cannot be reduced to overthinking
>
> ## **2️⃣ Additional Analyses Regarding Difficulty, Memorization, and Model Types**
>
> ---
>
> ### **🔷 Does Reasoning Rigidity Correlate With Difficulty?**
>
> We thank the reviewer for raising this important question.
>
> Our understanding of the reviewer’s point is:
>
> > *If reasoning rigidity becomes stronger on harder problems, perhaps the phenomenon partially stems from memorization or familiar templates, rather than true reasoning-level inflexibility.*
> Through our experiment, we prove that this is not the case, since reasoning rigidity is observed universally across all difficulty levels, from easy to hard. To investigate this, we follow the difficulty-evaluation methodology used in **SkyT1** and annotate both the *original* and *modified* ConditionedMath problems on a **1–10 difficulty scale**. We then plot pass@1 scores across AoPS difficulty levels in Figure 3 of the revised paper.
>
> **Findings**
>
> - **Rigidity exists consistently across all difficulty levels**.
>
> - Base models outperform reasoning models on modified problems **regardless of difficulty**, even though reasoning models outperform base models on the original tasks.
>
> - This indicates that rigidity is **not tied to task difficulty nor to memorized patterns**, but rather reflects a **content-level fixation** that persists across easy and hard problems alike.
>
> This supports our claim that reasoning rigidity is distinct from both memorization-driven errors and classical generalization failures.
>
> If the reviewer intended a different form of difficulty analysis (e.g., dynamic complexity, novelty metrics), we would be happy to run additional experiments.
>
> ## **3️⃣ Training-Time Mitigation and Future Work**
>
> ---
>
> ### **🔷 Feasibility of Training-Level Mitigation**
>
> We appreciate the reviewer’s suggestion regarding training-level approaches such as:
>
> - shorter or truncated reasoning traces (long-to-short CoT),
>
> - fine-tuning directly on ReasoningTrap.
>
> Due to labor intensive quality monitoring process of constructing ReasoningTrap, the dataset cannot be scaled up to large dataset to support supervised fine-tuning. Moreover, our primary focus in this work was on identifying and analyzing the previously under-explored reasoning rigidity phenomenon, and offer a simple yet effective **inference-time mitigation**.
>
> Nevertheless, to partially explore the reviewer’s suggestion, we plan to test long-to-short reasoning models in future work, which aim to reduce reliance on long-form CoT and encourage more concise internal solutions.

---

### Official Review · Reviewer_S4gG · 2025-11-01

**Soundness:** 3
**Presentation:** 2
**Contribution:** 3
**Rating:** 4
**Confidence:** 4

**Summary:**

Authors in this paper, discussed an interesting failure mode of "thinking" models or LRMs, namely reasoning rigidity --- where the model ignores problem-specific variations and instructions, and default to known problem-solution mappings. To study this, the authors propose ReasoningTrap, a diagnostic dataset including modified mathematical and logical problems (from AIME, MATH500, and puzzle sets) designed to test instruction adherence. They identify three rigidity modes: Interpretation Overload, Input Distrust, and Partial Instruction Attention, and propose some mitigation strategies. Empirical analysis across several reasoning models (Qwen, DeepSeek, GPT, Gemini, Claude) shows that reasoning-aligned models often perform worse than base models on the modified tasks (especially on known problems and their variant pairs), indicating a bias toward familiar solution paths.

**Strengths:**

1. The notion of reasoning rigidity is interesting. A better interpretation could have been drawn from CBR. Nonetheless, this seems to be an useful failure mode to analyse
2. Authors introduce a benchmark to study the failure mode comprising of both logical and math problems. This seems useful. ReasoningTrap seems to be carefully structured to reveal subtle reasoning biases and contamination effects.
3. Experiments show some evidence to the existence of the rigidity problem.

**Weaknesses:**

1. The phenomena, lies in the middle of instruction adherence and reasoning failure. It seems like a small subproblem. I am unsure how important or insightful it is to be studied separately.
2. Dataset creation overtly relies on GPTs.
3. Annotation procedures lack crucial details.
4. Certain findings (e.g., base models outperforming reasoning models) are neither deeply analyzed nor theoretically argued.
5. Some concerns with choices of the experimental setup (see below)

**Questions:**

L150: I am not sure how these two are the only two types. There are so many types of benchmarks, focusing on various logical languages, various levels of mathematics with increasing complexity.
In fact, other domains such as finance domain etc may have such examples as well, that could have supported the current (somewhat narrow) problem scope.

L183: I am somewhat unclear as to the motivation of using GPTs to generate modifications as well. Doesn't that inherit the bias within these specific models. Using GPT as verifier blindly without any further justification does not make sense either.
While the authors have done manual verification, I am not satisfied with the missing details about the annotation or verification process. This seem to have been trivialized.

L205: What I do not understand is the over-reliance of GPTs to generate datasets? Are we claiming that all other methods, manual methods, templated ones, or human-and-AI in the loop methods do not work well? There is not even a justification.

L322: While authors have adopted this experimental setup, I really do not know how relevant is this as a scientifc study. Observations such as this simply would entail that the base models have certain better abilities than their so-called "thinking" counterparts. This is nothing to be surprised about. With continual learning, forgetting is a well-observed phenomena.

Sec 5.2: What does this section 5.2 mean? By definition, the original problem should be "common"/"known" and the variant should be unknown. Otherwise, the variant problem's solution could be mapped as well. I do not understand what "unseen" is supposed to mean.

L405: abuctive -> deductive.

---

> ### Author Response · Authors · 2025-11-21
> **Author Response to Weaknesses - Part 1**
>
> ## **1️⃣ Distinguishing Reasoning Rigidity from Other Reasoning-Related Failures**
> ---
>
> | Failure Mode | Ability Present | Active Denial | Fixed Reasoning | Primary Symptom |
> | --- | --- | --- | --- | --- |
> | **Instruction Following** | ✗ | ✗ | ✗ | Cannot follow CoT *format* directives |
> | **Reasoning Faithfulness / Inability** | ✗ | ✗ | ✓ | Incorrect / faithless reasoning |
> | **Memorization** | ✗ | ✗ | ✓ | Failure to generalize |
> | **Overthinking** | ✓ | △ | △ | Unnecessarily long CoT |
> | **Reasoning Rigidity** | ✓ | ✓ | ✓ (content) | Refuses the user instruction to follow familiar reasoning |
>
> **Table:** Condensed comparison of reasoning-related failure modes. **Legend:** ✓ = strongly related; △ = partial; ✗ = unrelated.
>
> ### **🔷 Key Distinction Between Reasoning Rigidity and instruction adherence**
>
> Existing works that explore instruction following ability in reasoning are limited to **format** level conditions such as length, output format, and tone. This differs from our work since reasoning rigidity focuses on whether the model rigidly follows the wrong but familiar reasoning **content**.
>
> ### **🔷 Key Distinction Between Reasoning Rigidity and Reasoning Inability**
>
> Reasoning inability encompasses diverse problems of models that can reason. Therefore, reasoning rigidity is also categorized as one of reasoning inability problem. Existing reasoning inability problems are investigated in multiple facets. For example, the most related work would be reasoning faithfulness literature, which reveals that reasoning models sometimes exhibit discrepancies between their latent decision process and the generated Chain-of-Thought(CoT) explanation [1-3]. While reasoning faithfulness serves as a useful indicator of how reliably one can trust a model’s textual rationale, it does not directly assess whether the model is capable of solving the problem or whether it is actively rejecting a correct reasoning direction. Thus, *faithfulness concerns the alignment between internal and external reasoning traces, and is fundamentally distinct from reasoning rigidity*, which arises when a model could solve the problem yet persistently adheres to an inappropriate but familiar reasoning pattern.
>
> Another related literature focuses on whether a model can follow user-specified control signals over the reasoning process - for example, to reason concisely, adopt an alternative strategy, or invoke a specific tool (e.g. ability of LLMs to adapt to creative problem solving [4], generalization to unseen variants of math word problems [5]). In contrast, reasoning rigidity characterizes cases where the model correctly interprets both the problem and the instruction but nevertheless defaults to a familiar reasoning template, failing to adapt its reasoning strategy to the modified task. Unlike inability, which concerns missing capabilities, rigidity highlights an inflexibility in deploying existing capabilities even when the correct solution is within reach.
>
> - [1] Madsen et al., Are self-explanations from Large Language Models faithful?, ACL Findings 2024
>
> - [2] Matton et al., Walk the Talk? Measuring the Faithfulness of Large Language Model Explanations, ICLR 2025
>
> - [3] Chen et al., Reasoning Models Don't Always Say What They Think, Arxiv 2025
>
> - [4] Naeini et al., Large language models are fixated by red herrings: Exploring creative problem solving and einstellung effect using the only connect wall dataset, NeurIPS 2023
>
> - [5] Raiyan et al., Math Word Problem Solving by Generating Linguistic Variants of Problem Statements, 2023
>
> Moreover, to emphasize the novelty of reasoning rigidity, we have reorganized the related work section and added the **Table** in the revised paper, which systematically contrasts reasoning rigidity against five related failure modes - instruction adherence, reasoning faithfulness, reasoning inability, memorization, and overthinking.
>
> ### **2️⃣ Data Create Relies on Both Human Experts and LLMs**
> ---
> Our dataset create do not overly rely on GPTs, but also employs human annotations to monitor over dataset filtering process.
>
> **ConditionedMath**
>
> > While LLMs were used to generate initial variants, all 84 instances were **rigorously verified by human annotators**. This ensures that each modified problem is logically valid, correctly reflects the intended transformation, and remains solvable. Thus, the dataset does *not* fully rely on LLM generation.
>
> **PuzzleTrivial**
>
> > For **PuzzleTrivial**, the **core logical transformation** of each item was authored by **human annotators**. Only after establishing a reliable core modification did we apply lightweight augmentations involving *logic-invariant conditions*. These augmentations never alter the essential logical structure. We have clarified this process in the revision.

---

> ### Author Response · Authors · 2025-11-21
> **Author Response to Weaknesses - Part 2**
>
> ## **3️⃣ Details of Annotation Procedure**
>
> ---
>
> We have fully revised the dataset annotation process in Section 4 to facilitate the understanding on dataset construction process. The revised sections are highlighted in blue.
>
> **Stage 1: Original Question Modification**
>
> We use OpenAI gpt-4o-mini for LLM-based dataset generation and o4-mini for LLM-based dataset filtering, since filtering requires a more powerful language model as a verifier. After LLM-based filtering, the dataset are further filtered into smaller high-quality subset by human annotators.
>
> For generating novel conditions, we prompt the model to modify the original problem and solution into five distinct versions that meaningfully alter the problem's reasoning trajectory and eventually lead to a different answer. As this process proliferates the original dataset into five-fold larger dataset with unverified modifications, we proceed to dataset filtering process that incorporates monitoring from both strong LLM and human expert.
>
> **Stage 2: Dataset Filtering Pipeline**
> - *LLM Verification*: These modified questions are further validated on three critical criteria with OpenAI o4-mini model: (a) mathematical validity of the modified conditions to ensure that no internal contradictions exist, (b) divergence of the resulting solution from the original problem's solution, and (c) existence of solution. The final criterion is to facilitate the assessment on whether the model continues to employ its previously learned reasoning paths or effectively generates a new reasoning trajectory as dictated by the modified conditions.
>
> - *Human Verification*: Three annotators with mathematical expertise evaluated each question–solution pair using three criteria - valid, diverge, and solvable - each rated on a five-point Likert scale (1 = negative, 5 = positive). For each pair, the annotators recorded a score for all three criteria. We retained only those questions receiving a score of 5 on all three dimensions, ensuring an expert-quality dataset.
>
>
>
> ## **4️⃣ Our Paper Includes Extensive Analysis**
>
> ---
>
> We have analyzed diverse aspects of reasoning and base model behavior over our diagnostic dataset. **We list the experiments conducted for analysis purpose on reasoning rigidity below**, and we would be very happy to accommodate the reviewer’s suggestion on extending the analysis if there are any missing important analysis aspect.
>
> > 1. Contamination Comparison in Base vs Reasoning Models. (Table 2)
>     - Reasoning models are more prone to be contaminated by wrong but familiar reasoning traces.
> 2. Contamination Score vs. Model Accuracy (Figure 2)
>     - Contamination score serves as a indirect indicator of reasoning rigidity, as trying familiar reasoning approach is a legitimate behavior of chain of thought reasoning process.
> 3. Problem Understanding and Reasoning Rigidity (Table 3 p-score)
>     - Rigidity Patterns Emerge in Various LRMs, from Open-sourced to Proprietary Models.
> 4. Reasoning Rigidity Appears Across All Difficulty Levels. (Figure 3)
> 5. Task Familiarity leads to Reasoning Rigidity and OOD Tasks do not Manifest Rigidity. (Table 4)
> 6. Reasoning Rigidity Persists Beyond Memorization Effects. (Table 5)
> 7. Possible Causes of Reasoning Rigidity
>     7-1 Length (Table 6)
>     7-2 RL training (Table 7)
> 8. Appendix: reasoning effort (Table 13)
> 9. Appendix: various RL training objectives (Table 15)
> 10. Appendix: various model sizes (Table 14)

---

> ### Author Response · Authors · 2025-11-21
> **Author Response to Questions**
>
> ### 🟨 **Q1: Problem Scope Focuses on Reasoning Intensive Tasks**
> We do not claim that reasoning tasks are limited to only two tasks. However, reasoning models are widely evaluated upon reasoning intensive tasks such as mathematics and logical reasoning. Therefore, we choose these two tasks as representatives to show the reasoning rigidity phenomenon. We also demonstrate that deductive logical reasoning task is comparably safe from reasoning rigidity problem, therefore there is no need to construct separate dataset for such OOD domains. We also provide various levels of difficulty (Figure 3) and categories (Table 11) in mathematics to prove that our dataset covers a wide variety of problems in mathematical domain.
>
> ### 🟨 **Q2: GPTs cannot not Infiltrate Rigidity Bias into Dataset**
> The reviewer raised an important point. Reasoning rigidity appears only in reasoning-oriented models. In our pipeline, the LLM-based dataset construction phase relies on a **non-reasoning GPT model**, which is not susceptible to rigidity biases. In addition, while the dataset filtering phase does use a reasoning-enabled GPT, the prompts used for filtering are structurally far from any familiar reasoning trace that could trigger rigidity. Specifically, the filtering prompt asks the model to assess question quality along three dimensions - validity, solvability, and difference from the original. None of these tasks resemble a problem-solving scenario that would invoke a memorized reasoning template. **Therefore, reasoning rigidity should not manifest in this setting**.
>
> ### 🟨 **Q3: Over-reliance on GPTs: We use Human Annotation & Filtering with LLM**
>
> The reviewer’s claim that our dataset construction overly relies on GPTs sees to be far from the actual construction process.
>
> Our method is closer to Human-and-AI in the Loop, as it employs LLM-based dataset annotation first, LLM-based dataset filtering with three template criteria, and finally human expert dataset filtering process with the same template criteria to ensure the dataset quality.
> The detailed dataset construction process has been fully revised in the paper section 4. We hope this clarifies the our dataset construction process.
>
> their so-called "thinking" counterparts. This is nothing to be surprised about. With continual learning, forgetting is a well-observed phenomena.
>
> ### 🟨 **Q4: Distinction with Existing Works: Base Model Outperformance and Catastrophic Forgetting**
>
> The reviewer claims
>
> > **Base Model Outperforming Reasoning Model in Certain Aspects is a Well-known Problem. This is nothing to be surprised about.**.
>
> There are several works that discovers base models outperform reasoning models in special cases where the problems are simple that it do not require any reasoning. However, we show that base model outperforms reasoning models even in olympiad level mathematical questions that require a long reasoning trace in Figure 3 of the revised paper. If reasoning rigidity was a known problem, then base model outperforming reasoning model would have only been observed between level 1-2 which indicate elementary level mathematics. Moreover, we also show that in original MATH500 and AIME datasets, reasoning models outperform base models in all levels, but this ranking is reversed when the problem is subtly modified to require a completely different solution path. This experimental design elegantly shows that reasoning rigidity is difficulty agnostic, and it is indeed a novel phenomenon, unexplored from previous works.
>
> Moreover, the reviewer also claims that
>
> > **With continual learning, forgetting is a well-observed phenomena.**.
>
> **Our work largely differs from catastrophic forgetting.** Catastrophic forgetting focuses on the lack of capability of the models that are continually trained on different tasks. First of all, reasoning models are trained on similar tasks, which fortifies the reasoning ability and enhance the model performance across various benchmarks. Therefore, it is plausible to say that catastrophic forgetting is not an issue with reasoning models trained from base models. Second, catastrophic forgetting should manifest that a model itself do not possess a ability to solve the problem, but in reasoning rigidity cases, the models have the ability to solve the problems but actively overrides the given problem conditions to default to familiar reasoning trace. This is shown by high perception score of reasoning models compared to base counterparts in Table 3.
>
> ### 🟨 **Q5: Meaning of Unseen vs Seen / Familiar (Common) vs Unfamiliar (Unknown)**
> We have tested DeepSeek R1 (released before AIME25) on AIME25 (**unseen**) to test if it still shows rigidity. The experimental result shows that reasoning rigidity is still shown in AIME25, which are apparently unseen training dataset, from which we conclude that reasoning rigidity is not a mere memorization problem. **Therefore the original problem should be "common" but is "unseen" in this case.**

---

### Author Response · Authors · 2025-11-21
**Global Response Regarding Revised Manuscript**

We sincerely appreciate all reviewers for their constructive feedback and reviewing efforts, which have greatly helped us improve our work.
We have updated the revised manuscript incorporating the concerns raised by the reviewers. To avoid any potential confusion for readers, we have highlighted the edited parts in blue text color.

Once again, we deeply thank for your time, effort, and valuable insights. Thank you for helping us strengthen our work.

---

### Author Response · Authors · 2025-11-27

Dear Reviewers,

As we haven't received any feedback after the rebuttal submission, we would be happy to address any remaining questions. We look forward to your valuable feedback at your earliest convenience.

---

### Note · Authors · 2026-01-04

**Comment:**

We appreciate the reviewers' effort for providing constructive feedback. As we are planning to polish our paper based on the feedbacks, we withdraw from ICLR 2026 submission.

**Withdrawal Confirmation:**

I have read and agree with the venue's withdrawal policy on behalf of myself and my co-authors.